# Activation of the archaeal ion channel MthK is exquisitely regulated by temperature

Yihao Jiang[1,2], Vinay Idikuda[1,2], Sandipan Chowdhury[1,2], Baron Chanda[1,2]*

[1]Department of Anesthesiology, Washington University School of Medicine, St. Louis, United States; [2]Center for the Investigation of Membrane Excitability Diseases (CIMED), St. Louis, United States

**Abstract** Physiological response to thermal stimuli in mammals is mediated by a structurally diverse class of ion channels, many of which exhibit polymodal behavior. To probe the diversity of biophysical mechanisms of temperature-sensitivity, we characterized the temperature-dependent activation of MthK, a two transmembrane calcium-activated potassium channel from thermophilic archaebacteria. Our functional complementation studies show that these channels are more efficient at rescuing $K^+$ transport at 37°C than at 24°C. Electrophysiological activity of the purified MthK is extremely sensitive ($Q_{10} > 100$) to heating particularly at low-calcium concentrations whereas channels lacking the calcium-sensing RCK domain are practically insensitive. By analyzing single-channel activities at limiting calcium concentrations, we find that temperature alters the coupling between the cytoplasmic RCK domains and the pore domain. These findings reveal a hitherto unexplored mechanism of temperature-dependent regulation of ion channel gating and shed light on ancient origins of temperature-sensitivity.

*For correspondence:
bchanda@wustl.edu

## Introduction

Ion channels act both as gatekeepers and signal transducers responding to a variety of environmental cues including both physical and chemical stimuli. Although most biological molecules respond to acute physical forces such as temperature or mechanical stretch, specialized ion channels are about an order of magnitude more sensitive than most ion channels. For instance, the $Q_{10}$ (fold change in activity for a 10°C change in temperature) is higher than 20 for a bonafide temperature-sensitive ion channel such as TRPV1 channel compared to $Q_{10}$ of 2–4 for most ion channels (*Clapham and Miller, 2011*; *Islas and Qin, 2014*). In addition to TRPV1 (*Caterina et al., 1997*), the activity of TRPM8 (*McKemy et al., 2002*), TREK (*Maingret et al., 2000*), TMEM16A (*Cho et al., 2012*) and even prokaryotic voltage-gated sodium channels (*Dib-Hajj et al., 2008*) are all highly regulated by temperature. As far as we know, these channels lack a common structural motif for sensing temperature, unlike their ligand activated counterparts. This lack of conserved structural module is not surprising given that sensors of physical stimuli are not constrained by a specific domain in contrast to chemical sensors (*Kuriyan et al., 2012*; *Goldschen-Ohm and Chanda, 2017*). In fact, it is unclear whether the force-sensing mechanism involves a discrete module or distributed microsensors (*Clapham and Miller, 2011*; *Islas and Qin, 2014*; *Chowdhury et al., 2014*; *Arrigoni et al., 2016*; *Arrigoni and Minor, 2018*; *Brauchi et al., 2006*; *Diaz-Franulic et al., 2016*; *Jabba et al., 2014*; *Raddatz et al., 2014*; *Yao et al., 2011*; *Grandl et al., 2008*; *Grandl et al., 2010*; *Yang et al., 2010*; *Zhang et al., 2018*). Thus, from a mechanistic standpoint, these physical force-sensing ion channels are referred to as Type III channels to distinguish them from ligand-activated ion channels which typically harbor conserved structural motifs (*Goldschen-Ohm and Chanda, 2017*).

From a biophysical perspective, most studies on mechanisms of temperature-dependent gating have focused on eukaryotic channels. These channels are large polymodal allosteric systems and many exhibit complex behavior such as irreversible gating which makes detailed thermodynamic analysis non-trivial (*Liu et al., 2011*; *Sánchez-Moreno et al., 2018*). Prokaryotic ion channels are widely used as model systems to probe basic mechanisms that underlie gating behavior and transport characteristics of their eukaryotic counterparts. These ion channels are more tractable to structural and biochemical analyses. For instance, much of our understanding of ion selectivity comes from studies on bacterial KcsA (*Doyle et al., 1998*) and NaK (*Shi et al., 2006*) channels. KcsA ion channels have also become exemplars to understand the structural mechanisms that underlie C-type inactivation (*Cordero-Morales et al., 2011a*; *Cuello et al., 2010a*). Prokaryotes, particularly archaebacteria, have been found in a variety of habitats, including in extreme environments such as hot springs and deep ocean vents. Although prokaryotic ion channels that sense physical stimuli such as light, voltage, and stretch have been well-established, very few studies have examined the temperature-sensitive gating in prokaryotic ion channels (*Arrigoni et al., 2016*).

In this study, we characterized the effect of temperature on gating of MthK, an archaebacterial calcium-activated potassium channel from *Methanobacterium thermoautotrophicum*, a thermophile abundant in the hot springs of Yellowstone National park (*Sandbeck and Ward, 1982*; *Jiang et al., 2002*; *Zeikus and Wolfe, 1972*). MthK is a tetramer with two transmembrane helices per subunit which come together to form a central pore (*Jiang et al., 2002*). The gating state of the central pore is regulated by RCK domains on the C-terminus of each subunit which undergo a conformational change upon binding to calcium and other divalent cations (*Smith et al., 2012*; *Pau et al., 2011*; *Dvir et al., 2010*). Parfenova and colleagues have previously suggested that temperature regulates the activity of MthK (*Parfenova et al., 2006*). Here, using an N-type inactivation-deficient variant of MthK (*Kuo et al., 2008*; *Fan et al., 2020*), we find that the electrophysiological activity of these channels is highly sensitive to temperature in both native *E. coli* membranes and purified preparations. Our studies reveal that the RCK domain is essential for temperature-dependent gating and the primary effect of temperature is to alter the coupling strength between the calcium-sensing RCK domain and pore domain.

## Results

### MthK expressed in *E. coli* is temperature-sensitive

To probe the temperature sensitivity of MthK in bacteria, we utilized a complementation assay based on *E. coli* potassium-uptake deficient LB2003 strain (*Stumpe and Bakker, 1997*). Expression of functional potassium transporters or channels rescues the growth of this strain under low-potassium conditions (*Parfenova et al., 2006*; *Hänelt et al., 2010*). Previous studies have shown that increased rescue of LB2003 strain can be directly correlated with a higher open probability of the expressed potassium channels (*Cuello et al., 2010b*). Here, we tested three MthK constructs - the full-length channel (MthK FL), an N-terminal deletion construct, which removes fast inactivation (MthK IR) (*Kuo et al., 2008*), and a truncated construct lacking the RCK domain (MthK ΔC) (*Figure 1A*) – for their ability to rescue growth of LB2003 strain at various temperatures. All the genes are under the control of lacY-inducible expression system.

Since bacterial growth is slower at low temperatures, it was necessary to incubate growth plates much longer at those temperatures. We also find that the apparent viability of the host LB2003 strain increases at lower temperatures presumably due to longer incubation times. Therefore, to normalize for these differences in growth rates and viability at different temperatures, we compared bacterial growth with and without induction for all constructs at the four set temperatures (*Figure 1B*). Compared to the uninduced controls, expression of MthK FL, MthK IR, and MthK ΔC all rescue the growth of LB2003 in low potassium plates at 24°C and 28°C, which suggests they can form a viable potassium channel in the membrane. The expression of MthK FL and MthK IR rescues cell growth more efficiently at 36°C than at 18°C compared to empty vector controls or even expression of MthK ΔC. To better quantify the differences in complementation between various constructs, we used a suspension culture assay. With overnight (~15 hr) incubation, transformed LB2003 cells reached the stationary phase at both 37°C and 24°C. The cell densities of these stationary phase reflect how well each construct rescued the LB2003 cell growth at the corresponding temperatures. As shown in

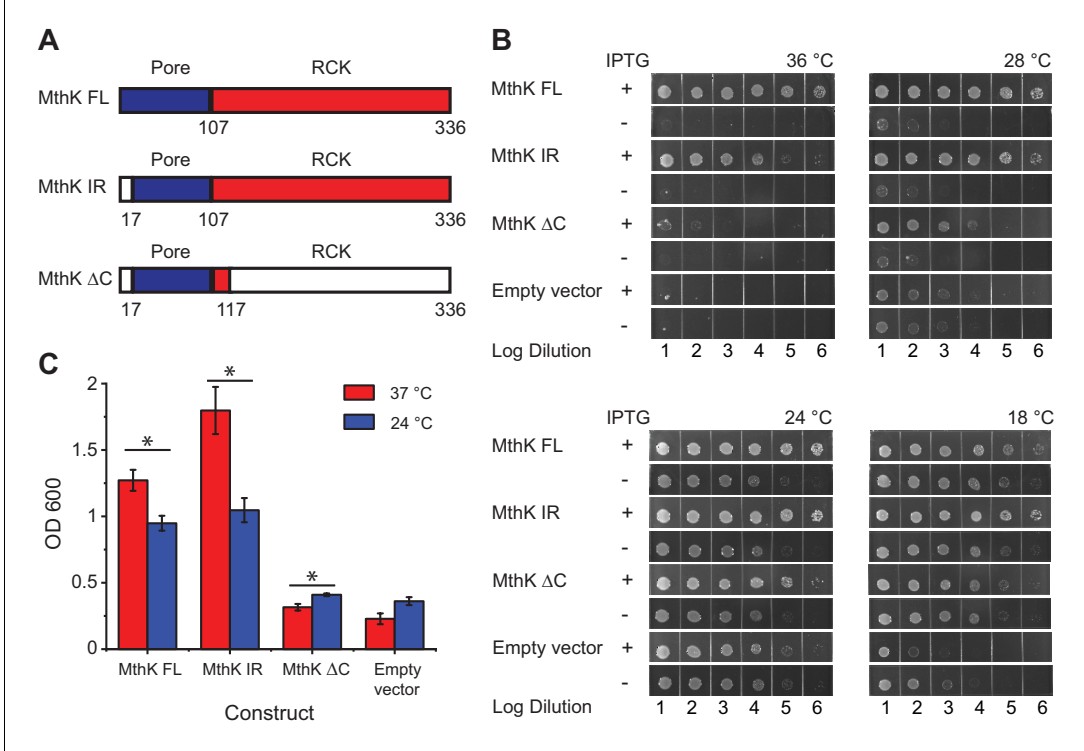

**Figure 1.** Temperature dependence of complementation of *E. coli* LB2003 strain by various MthK constructs. (**A**) Schematic outlining the three different MthK constructs used in this study. The first 107 amino acids, which include the short cytoplasmic N-terminus (residues 2–17) and the two transmembrane helices are highlighted in deep blue. The C-terminus cytoplasmic region containing the RCK domain is shown in red. (**B**) Panels showing complementation of LB2003 by MthK constructs at different temperatures. Each column represents $\log_{10}$ dilution as indicated below. The plates were incubated for 18 hr at 36°C, 24 hr at 28°C, 48 hr at 24°C, and 96 hr at 18°C in order to compensate for differences in growth rate (see Materials and methods) (**C**) Complementation of LB2003 by MthK constructs at 37°C and 24°C in mini suspension cultures. Error bars represent SEM from n = 4 (MthK FL, 37°C), 3 (MthK FL, 24°C), 4 (MthK IR, 37°C and 24°C), 3 (MthK ΔC 37°C and 24°C), 3 (empty vector, 37°C), and 5 (empty vector, 24°C) independent experiments. Significance was determined by Student's two-sample T-test (* for p<0.05; ** for p<0.01).

The online version of this article includes the following source data and figure supplement(s) for figure 1:

**Source data 1.** Source data for *Figure 1C* and student's two sample t-test analysis.
**Figure supplement 1.** Complementation of LB2003 with MthK constructs at 37°C and 24°C.
**Figure supplement 1—source data 1.** OD600 bacterial density measurements and student's two sample t-test analysis.
**Figure supplement 2.** MthK IR expressed in bacterial membranes shows increased activity at higher temperatures.
**Figure supplement 2—source data 1.** Source data for Fig. 1-S2D normalized macroscopic currents.

*Figure 1C*, similar to the plate assay, both MthK FL and MthK IR rescued the bacterial growth better at 37°C compared to 24°C, consistent with a heat-activated phenotype of these channels. However, in the suspension culture assay, MthK IR showed a more dramatic temperature-dependent complementation compared to MthK FL, this could be a result of the lack of N-type inactivation in MthK IR construct. Furthermore, we used 5 mM $Ba^{2+}$ to block the MthK channel to ensure that the effects we observe are not from alternative mechanisms unrelated to $K^+$ permeation from MthK channel. MthK FL and MthK IR transformed bacteria rescue growth better in the absence of the blocker, at both 37°C and 24°C, while bacteria transformed with MthK ΔC and empty vectors show no difference (*Figure 1—figure supplement 1A and B*). These findings suggest that the cytosolic C-terminal domain of MthK is important for a robust temperature-dependent complementation. Surprisingly, we also find that MthK ΔC has a slightly increased survivability at 24°C compared to 37°C (*Figure 1C*). Similar temperature dependence is also observed in the presence of the blocker (*Figure 1—figure supplement 1C*) and therefore likely reflects better survivability of LB2003 at lower temperatures.

To assay the function of MthK in bacterial membranes, we generated giant *E. coli* spheroplasts (*Kuo et al., 2007*; *Martinac et al., 2013*; *Kikuchi et al., 2015*) to measure macroscopic currents responses. For these experiments, we used the N-type inactivation removed construct (MthK IR)

because the presence of inactivation obfuscates the interpretation of the equilibrium gating properties (*Liu et al., 2011*). Inside-out patch-clamp recordings in the presence of 1.5 mM calcium (close to the $EC_{50}$ [*Pau et al., 2010*]) show that the macroscopic currents increase at higher temperatures (*Figure 1—figure supplement 2A*). These macroscopic currents are contaminated by leak currents, which are also likely to be slightly temperature-dependent. We measured the leak currents at three different temperatures after blocking the MthK currents with 0.3 mM internal barium (*Figure 1—figure supplement 2B*; *Thomson and Rothberg, 2010*). Comparison of currents with and without barium shows very little baseline activity at low temperatures but at higher temperatures especially at 41°C, robust MthK currents are observed. The fold increase in temperature-dependent currents becomes evident after subtracting the baseline leak (*Figure 1—figure supplement 2D*). Precise estimates of $Q_{10}$ is not possible from these measurements because we cannot reliably estimate the small currents at low temperatures to calculate the fold change in activity as a function of temperature. Moreover, it is likely that our macroscopic current measurements underestimate the real change in the current amplitudes because barium block appears to be incomplete at higher temperatures (*Figure 1—figure supplement 2B*). A better approach to address these concerns would be to measure the temperature-dependence of MthK open probability directly by single-channel recordings. Furthermore, single-channel recordings at different temperatures allow us to determine whether the observed increase in macroscopic currents (*Figure 1—figure supplement 2C*) is due to increased open probability or increased single-channel conductance.

## Intrinsic temperature dependence of purified and reconstituted MthK

We sought to determine whether the observed temperature-sensitivity of MthK gating is intrinsic to the protein or is mediated by additional cellular cofactors. MthK IR was purified using metal affinity and size exclusion chromatography (SEC) from *E. coli* expression system as described previously (*Jiang et al., 2002*). The SEC elution profile and SDS-PAGE analysis of purified protein are shown in *Figure 2—figure supplement 1A* and *Figure 2—figure supplement 1B*. The purified MthK protein was reconstituted into soybean polar lipid vesicles. Giant multilamellar vesicles were generated and currents were recorded using inside-out patch electrophysiology as described previously (*Chakrapani et al., 2007*).

Single-channel recordings from the same patch at different temperatures in the presence of 0.1 mM calcium are shown in *Figure 2A* and *Figure 2—figure supplement 1C*. Large conductance of the open MthK channel allows us to resolve opening events easily. Up until 32°C, single-channel openings are rare, but upon elevating the temperature to 37°C, we observe a large increase in single-channel activity. The current amplitude distributions at various temperatures clearly show that the channels remain closed till 32°C, but upon a further increase in temperature, opening events increase dramatically (*Figure 2B*). The open-dwell time distributions clearly indicate the presence of multiple kinetically distinct open states (*Figure 2—figure supplement 2*). There is no significant change in the mean open-dwell time between 21°C and 37°C (*Figure 2C*). However, the open probability, Po, calculated from 5 min of continuous single-channel recordings, shows that at 37°C, Po is about 20 times higher than those measured at lower temperatures (*Figure 2D*).

As

$$mean\ closed\ time = (mean\ open\ time/\mathrm{Po}) - mean\ open\ time, \tag{1}$$

The mean close-dwell time changes from ~930 ms at 21°C to ~44 ms at 37°C. Thus, the high temperature profoundly shortens the residence times of the channel in the closed states which would dramatically reduce the stability of closed states and account for the change in Po.

The observed temperature dependence of the single-channel activity of purified MthK reconstituted in soybean lipids corresponds to a $Q_{10}$ of over 100, which is comparable to the canonical eukaryotic temperature-sensitive ion channels such as those of the TRP family of channels. Our findings also establish that temperature-sensitive activation is an intrinsic property of the channel and does not require additional cellular cofactors.

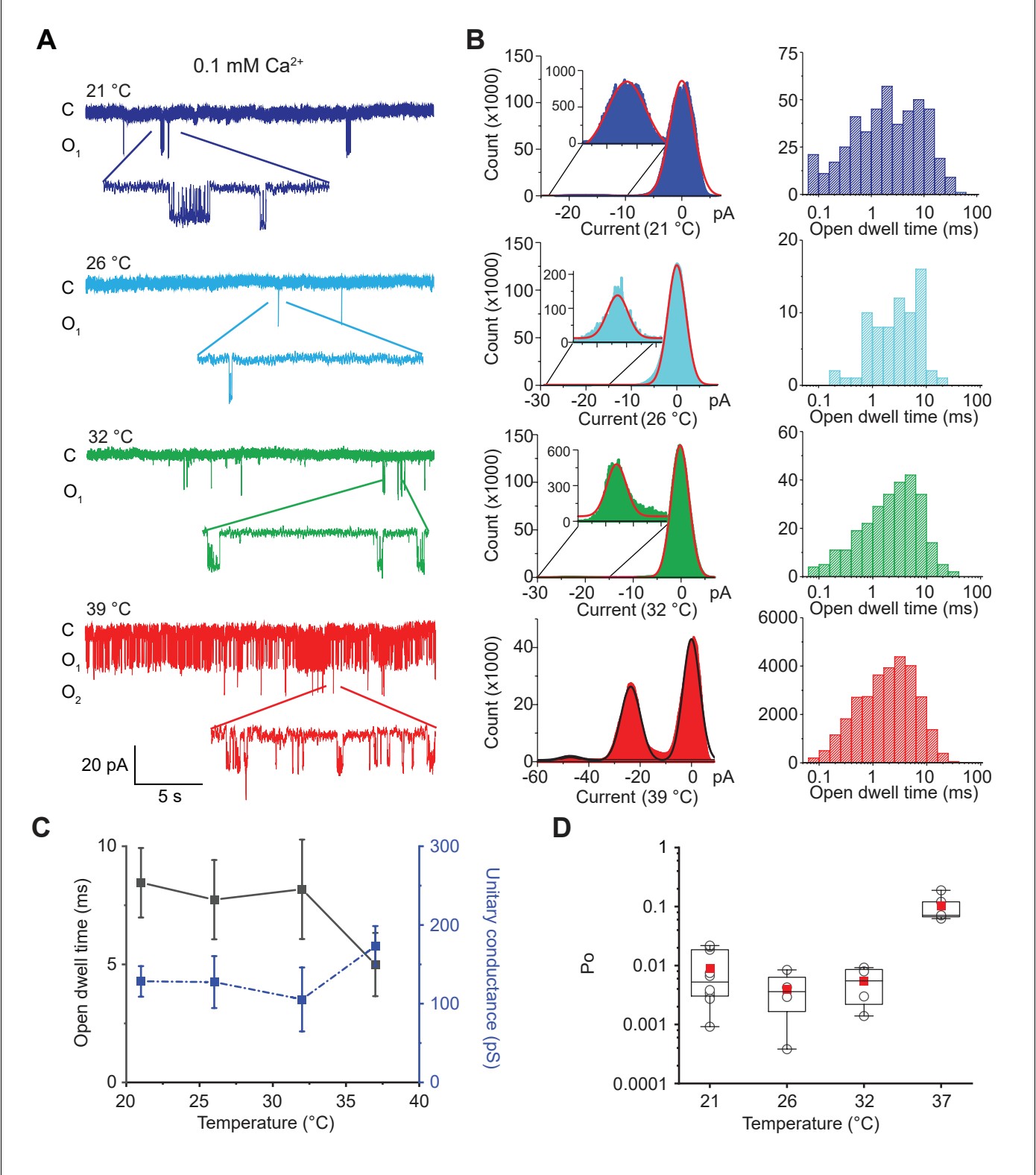

**Figure 2.** MthK IR is intrinsically heat sensitive. (**A**) Representative single-channel recordings of purified MthK IR reconstituted in soybean polar lipids with 0.1 mM calcium at various temperatures. All traces were recorded at −100 mV from the same patch. (**B**) Current amplitude histograms (left) and open-dwell time (right) of single-channel recordings with the same color scheme as in (**A**). Inset: The current amplitude histograms corresponding to open channels are enlarged for clarity. (**C**) Open-dwell time (solid black line) and unitary conductance (dashed blue line) of MthK IR plotted with respect

*Figure 2 continued on next page*

*Figure 2 continued*

to temperature. (D) Box graph of open probability at various temperatures. The boxes indicate 25–75 percentiles of the data; the whiskers show 5–95 percentiles of the data. Within each box, the line indicates the median value and red symbol marks the mean. Each individual data point is shown as circles. Error bars for (C) are SEM calculated from n = 10 (21°C), 4 (26°C), 4 (32°C), 5 (37°C) independent measurements.

The online version of this article includes the following source data and figure supplement(s) for figure 2:

**Source data 1.** Source data for each open probability, open dwell time and condctance data point in the presence of 0.1 mM Ca²⁺.
**Figure supplement 1.** Purification and single-channel recordings of MthK IR.
**Figure supplement 1—source data 1.** Source data for gel filtration profile of MthK IR.
**Figure supplement 2.** Open-dwell time histogram fitted with exponentials.
**Figure supplement 2—source data 1.** Source data for bin plot of open dwell time.
**Figure supplement 3.** Heat activation of MthK channel is reversible.
**Figure supplement 3—source data 1.** Source data for reversibility of MthK IR.

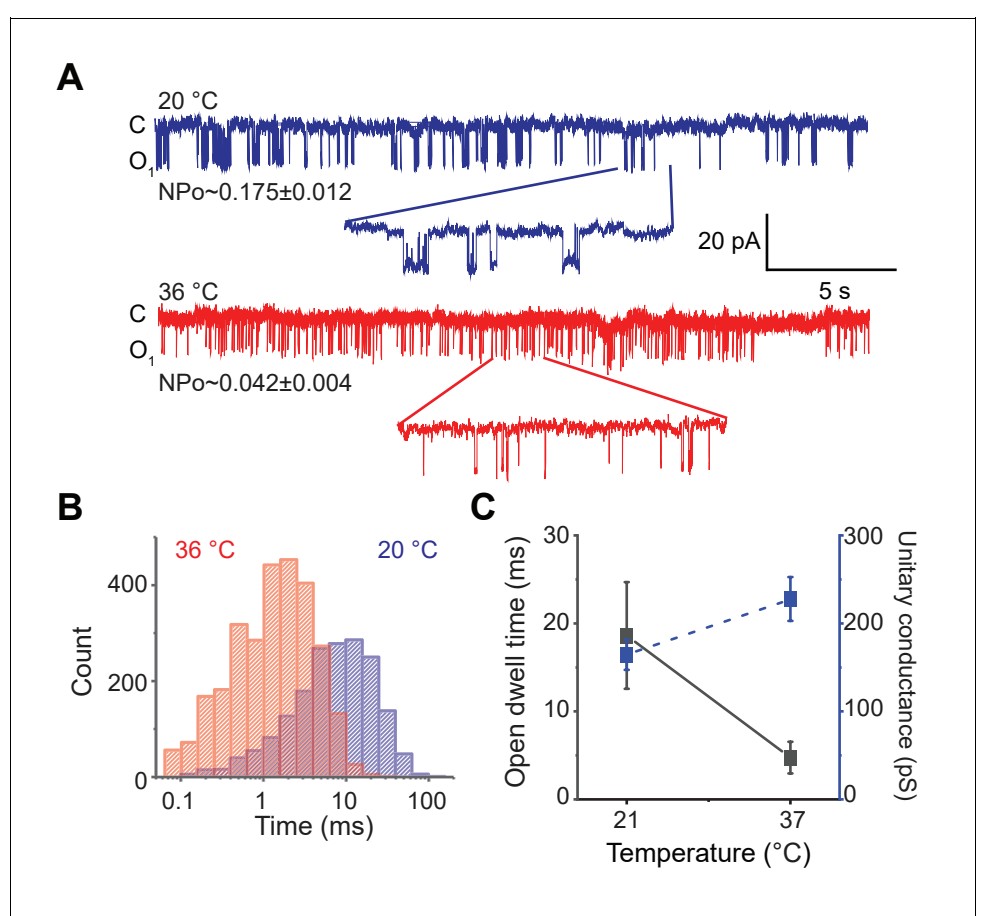

**Figure 3.** Ablation of RCK domain reduces the temperature-dependence of MthK. (A) Representative single-channel recordings of purified MthK PO reconstituted in soybean polar lipids with 1 mM calcium at 20°C (blue) and 36°C (red). Inset shows the enlarged traces. nPo values are shown below each trace; the errors were calculated as SEM from bootstrapping the whole recordings. (B) Open-dwell time histograms at the two different temperatures. (C) Plot of open-dwell times (solid black) and unitary conductance (dashed blue) of MthK PO with respect to temperature. Error bars represents SEM calculated from n = 5 independent patches.

The online version of this article includes the following source data and figure supplement(s) for figure 3:

**Source data 1.** Source data for *Figure 3B and C*.
**Figure supplement 1.** Purification and single-channel recordings of MthK PO.
**Figure supplement 1—source data 1.** Source data for gel filtration profile comparision of MthK IR and MthK Pore.

## The MthK pore domain is not responsible for temperature-dependent response

Calcium-dependent gating of MthK channel has been described via an allosteric scheme where the pore can exist in either closed or open state with an intrinsic bias towards the closed state. Upon calcium-binding shifts this bias toward the open state (*Jiang et al., 2002*; *Li et al., 2007*; *Lewis and Lu, 2019*). Our single-channel measurements were performed at very low calcium concentrations and so we wondered whether temperature skews the innate conformational bias of the pore domain. Such a hypothesis would parallel a recent study which showed that the pore domain of the heat-sensitive TRPV1 channel when fused to the voltage-sensing domain of the prototypical Shaker potassium channel, results in a chimeric channel with strong heat sensitivity, suggesting that pore domain of TRPV1 might retain the essential structural elements for temperature sensitivity (*Zhang et al., 2018*).

To directly test the temperature sensitivity of the pore domain, we adapted a previously reported protocol to purify a 'pore-only' MthK channel (MthK PO) (*Li et al., 2007*; *Posson et al., 2013*). Purified MthK IR was treated with trypsin to cleave off the RCK domains, and the 'pore-only' domain was subsequently isolated via gel-filtration chromatography (*Figure 3—figure supplement 1A and B*). MthK PO was reconstituted into soybean polar lipid vesicles and its single-channel activity was measured at 20°C and 36°C (*Figure 3A* and *Figure 3—figure supplement 1C*). Although more opening events are observed at 36°C, the mean open-dwell times become shorter at high temperature compared to room temperature (*Figure 3B and C*). The mean open-dwell time at 21°C corresponds to 19 ms, whereas at 37°C, it corresponds to 5 ms. This ~4-fold change in mean open-dwell time of Mthk PO almost entirely accounts for 4.8-fold higher nPo near room temperature. These effects are in stark contrast to those observed in MthK IR where the open probability increases by about two orders of magnitude upon heating. Taken together, these results strongly suggest that opening of the isolated pore domain of MthK is not temperature-sensitive and that structural elements from RCK domains are required for its exquisite heat-sensitivity.

## Coupling of RCK domains with the pore

Although the canonical heat-sensitive TRPV1 channels has been used to probe the thermodynamics of temperature-sensitivity, recent studies have shown that these channels exhibit hysteresis and irreversible loss of activity upon repeated stimulation (*Sánchez-Moreno et al., 2018*). Other studies have also shown that persistent heat causes rapid desensitization of the wild-type TRPV1 channel (*Luo et al., 2019*; *Cui et al., 2012*). For MthK IR, we have not observed any significant temperature-dependent desensitization when patches are held at high temperatures (39°C) up to 5 min (*Figure 2—figure supplement 1*). Furthermore, the Po values of MthK IR at room temperature are similar to those obtained after heat activation (*Figure 2—figure supplement 3*) indicating that the temperature-dependent gating is reversible (between 21°C and 37°C) and that MthK IR may serve as a suitable model for further thermodynamic analysis.

To probe the effect of temperature on allosteric regulation of pore gating, we measured single-channel activities of MthK IR at different calcium concentrations at two temperatures, 21°C and 37°C (*Figure 4A and B*). The calcium dose-response curves of MthK obtained from Po estimates exhibit high Hill-coefficients (*Zadek and Nimigean, 2006*), which makes it very challenging to obtain accurate measurements of $EC_{50}$. Nevertheless, the saturating regimes of calcium concentrations constrain these values within a specific range. Our measured dose-response curves of MthK reveal that the $EC_{50}$ of calcium at 21°C and 37°C are only modestly different ($EC_{50}$ ~0.7 mM at 21°C versus 0.6 mM at 37°C), although there is a decrease in the steepness of the dose-response curve (4.0 at 21°C versus 1.8 at 37°C) (*Figure 4—figure supplement 1A and B*). Interestingly, much higher temperature dependence of MthK activity was observed at the lowest concentrations of calcium, where the RCK domains are primarily in apo state (*Figure 4C*). The lowest calcium concentration tested was 0.1 mM because patches became unstable below this concentration even in the presence of millimolar concentrations of magnesium, as has been reported previously (*Coronado, 1985*; *Graber et al., 2017*). In the most parsimonious allosteric model of calcium gating, the apo RCK domain does not interact with the pore but in the presence of calcium, it stabilizes the open pore resulting in increased Po. However, given our observations that robust thermosensitivity of MthK requires intact RCK domains and low occupancy of calcium-binding sites, we conclude that the apo-RCK domain regulates

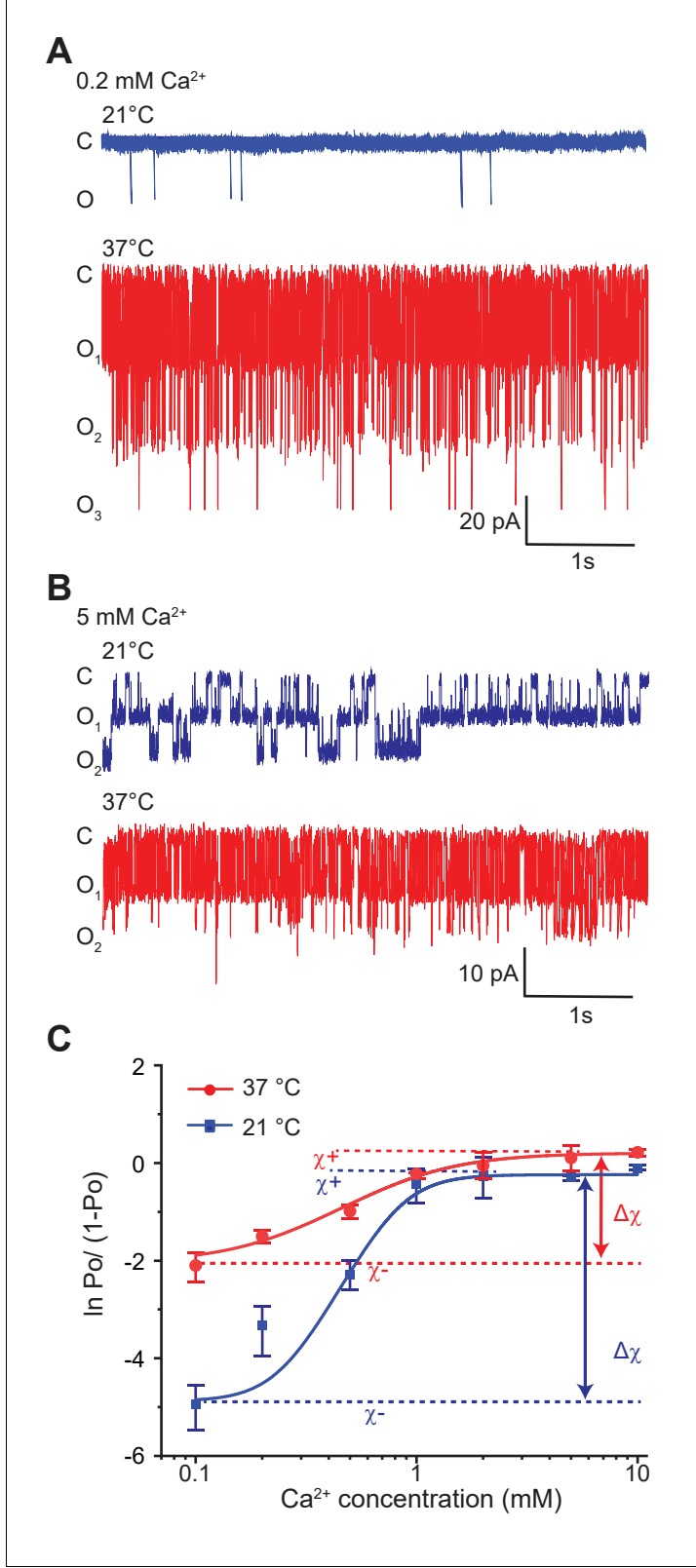

**Figure 4.** Temperature alters the coupling energy between the RCK and pore domain. Representative single-channel recordings in the presence of 0.2 mM calcium (**A**) and 5 mM calcium (**B**) at two different temperatures. For a particular calcium concentration, the recordings were obtained from the same patch. (**C**) Hill plot of ln [*Po/(1-Po)*] versus calcium concentration. Low-temperature data are shown in blue while high-temperature data are in red.

*Figure 4 continued on next page*

*Figure 4 continued*
The limiting asymptotes for each curve are depicted as dotted lines. $L_0$ values at each temperature correspond to the difference between asymptotes at those temperatures (indicated by double-headed arrows). Error bars represent SEM calculated from n = 10 (0.1 mM $Ca^{2+}$, 21°C), 3 (0.2 mM $Ca^{2+}$, 21°C), 5 (0.5 mM $Ca^{2+}$, 21°C), 6 (1 mM $Ca^{2+}$, 21°C), 4 (2 mM $Ca^{2+}$, 21°C), 4 (5 mM $Ca^{2+}$, 21°C), 5 (10 mM $Ca^{2+}$, 21°C), 4 (0.1 mM $Ca^{2+}$, 37°C), 3 (0.2 mM $Ca^{2+}$, 37°C), 4 (0.5 mM $Ca^{2+}$, 37°C), 3 (1 mM $Ca^{2+}$, 37°C), 4 (2 mM $Ca^{2+}$, 37°C), 3 (5 mM $Ca^{2+}$, 37°C), and 3 (10 mM $Ca^{2+}$, 37°C) independent recordings.
The online version of this article includes the following source data and figure supplement(s) for figure 4:

**Source data 1.** Source data for *Figure 4C*.
**Figure supplement 1.** Hill-plots of the calcium dose-response curves.
**Figure supplement 1—source data 1.** Source data for calcium activation curve at 21°C and 37°C.

temperature-dependent pore gating. Our findings also imply that apo-RCK domain must interact with the pore, consistent with a recent allosteric model (*Lewis and Lu, 2019*) but in contrast to earlier simple models (*Li et al., 2007*).

To determine whether the effect of temperature is on coupling interactions or on intrinsic equilibrium constants, we turn to linkage analysis (*Chowdhury and Chanda, 2012*; *Sigg, 2013*). We use the binary elements representation of allosteric models (*Goldschen-Ohm et al., 2014*), in which the pore and the calcium sensor each exist in two different conformations, analogous to the classical model. The transition of the pore, from the closed to open state, has an intrinsic equilibrium constant and the binding of calcium to the sensor is associated with an intrinsic binding affinity. However, instead of using a single coupling constant to describe the interactions between the pore and the calcium sensor, we use four different state-dependent interaction terms (*Figure 5A*). The classical models of allostery are a simplified version of this model with the different state-dependent interaction terms buried within the apparent equilibrium and coupling constants (*Chowdhury and Chanda, 2010*) (see 'Linkage analysis of Generalized Allosteric models' in Materials and methods section).

In the context of allosteric models, the net coupling energy between the sensor and pore domains can be calculated directly by measuring the difference between Hill transformed Po values in the presence and absence of stimulus as described previously (*Sigg, 2013*; *Chowdhury and Chanda, 2010*). This approach is analogous to the Hill-plot analysis of ligand-binding curves described by Wyman in his classical descriptions of allosteric linkage (*Wyman, 1967*). Here, we have outlined this approach for deconstructing the binding gating problem in ligand activation pathway without fitting the data to a specific allosteric model (see 'Linkage analysis of Generalized Allosteric models' in Materials and methods section).

In *Figure 4C*, the Hill-transformed Po (i.e. ln [$Po$/($1$-$Po$)]) with calcium concentration were plotted at 21°C and 37°C. In physical terms, the difference between the two asymptotes shown in *Figure 4C*, $\Delta\chi$ (at a specific temperature), may be expressed as:

$$\Delta\chi = \ln \sum \frac{\theta_{BO}\theta_{UC}}{\theta_{BC}\theta_{UO}} \tag{2}$$

where $-RT ln\theta_{BO}$ and $-RT ln\theta_{BC}$ are the interaction energies between a calcium-bound site with open-pore and closed-pore, respectively; while $-RT ln\theta_{UO}$ and $-RT ln\theta_{UC}$ are the interaction energies between an apo binding site with open-pore and closed-pore, respectively (*Figure 5A*). Thus, $\Delta\chi$ reflects the preference of the calcium-binding sites and the pore for 'like' conformations (i.e. bound-open or unbound-closed) as opposed to 'unlike' conformations (i.e. bound-closed or unbound-open).

Strikingly, Hill-plots of MthK IR at the two temperatures show that $\Delta\chi$ is different at the two temperatures. Going from 21°C to 37°C, $\Delta\chi$ changes from 4.5 to 2.2, which corresponds to a change in coupling energy from 2.6 to 1.4 kcal/mol. The experimental challenge associated with measurement of small Po values at low calcium concentrations raises some uncertainty about the exact magnitude of change in $\chi_-$ and thus $\Delta\chi$. We note that the Po measurements at 0.1 mM Ca is well-constrained by large number of independent replicates. It is possible that if the calcium concentration is lowered further, ln[$Po$/($1$-$Po$)] value may go down but this would only mean that the coupling energy $(\Delta\chi)$ at

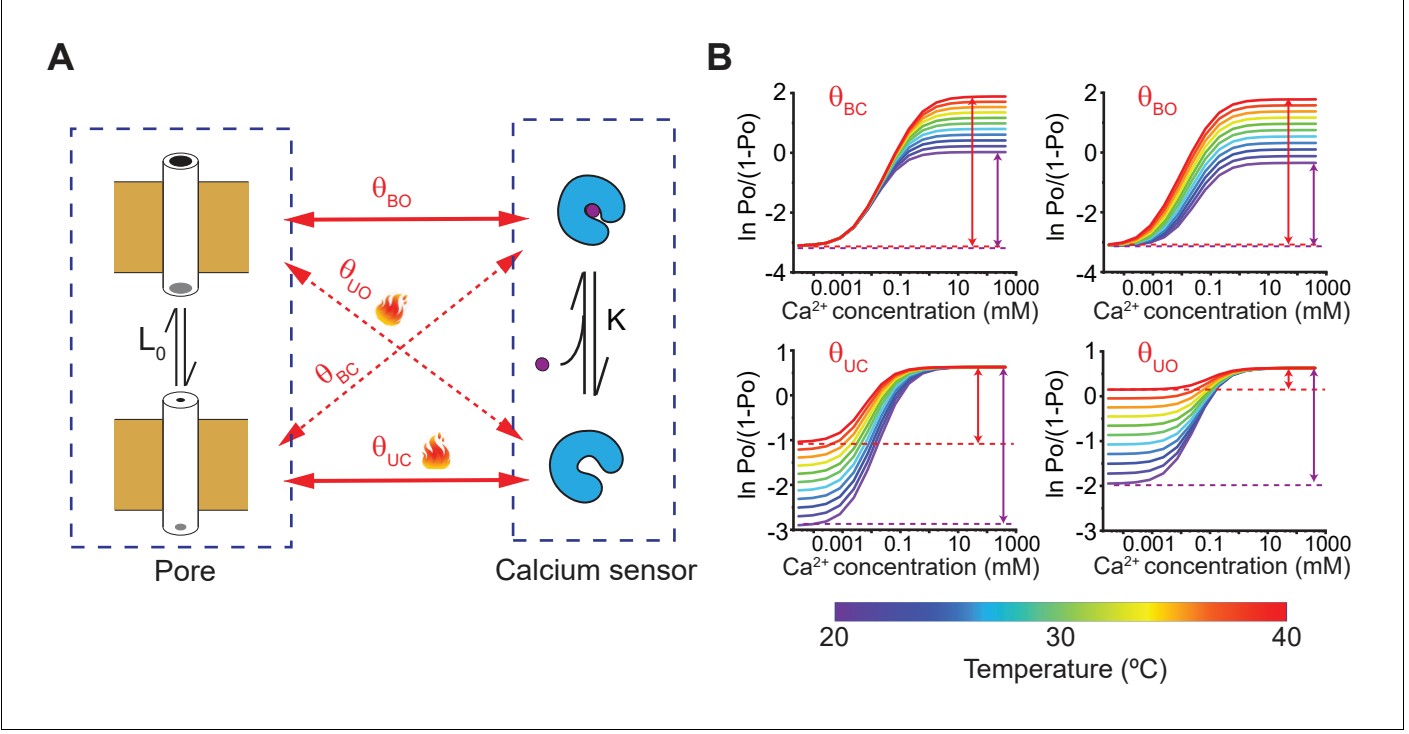

**Figure 5.** A simple allosteric model for temperature activation of MthK. (**A**) Four state binary elements model of MthK activation. The intrinsic equilibrium constant of pore opening is $L_0$ and the calcium-binding affinity is $K$. The state-dependent interactions between the pore and calcium sensor are represented by $\theta_{XY}$, where $X$ indicates the conformation of the calcium sensor ($X$ = bound (B) or unbound (U)) and $Y$ indicates the conformation of the Pore ($Y$ = open (O) or closed (C)). Solid lines indicate the interactions between 'like' states and the dotted lines highlight interactions between 'unlike' states. (**B**) Using the model of MthK channel gating, described in A, we simulated the Hill-plot of MthK (i.e. ln [$Po/(1-Po)$] vs. $Ca^{2+}$) at different temperatures for various model parameters. In each graph, all parameters, except the state-dependent interaction terms shown in each sub-panel, were kept constant across different temperatures. For all simulations, the values of the parameters used were: $L_0$ = 0.1; K = 20000 $M^{-1}$; $\theta_{UO}$ = 8; $\theta_{UC}$ = 18; $\theta_{BO}$ = 150; $\theta_{BC}$ = 8. For temperature-dependent simulations of $\theta_{BC}$, we use the following equation: $\theta_{BC}(T) = \exp(-(\Delta H_{BC} - T\Delta S_{BC})/RT)$, where $H_{BC}$ = -71 kJ and $S_{BC}$ = -220 J/K. Similarly, $\theta_{BO}$ ($H_{BO}$ = 81 kJ and $S_{BO}$ = 310 J/K), $\theta_{UC}$ ($H_{UC}$ = -71 kJ and $S_{UC}$ = -220 J/K) and $\theta_{UO}$ ($H_{UO}$ = 80 kJ and $S_{UO}$ = 300 J/K) were calculated.

The online version of this article includes the following source data and figure supplement(s) for figure 5:

**Source data 1.** Source data for simulated hill plot for various coupling parameters.

**Figure supplement 1.** Effect of temperature-dependent parameters on simulated Hill-plots of calcium-dependent gating of MthK.

**Figure supplement 1—source data 1.** Source data for simulated hill plot for pore intrinsic equilibrium $L_0$ and binding affinity KB.

**Figure supplement 2.** MthK activation involving multiple calcium binding sites with differing temperature-dependence.

**Figure supplement 2—source data 1.** Source data for simulated hill plot for multiple temperature binding sites with various temperature dependence.

**Figure supplement 3.** An alternate allosteric model of MthK activation involving independent calcium and temperature sensing domains.

**Figure supplement 3—source data 1.** Source data for simulated hill plot for a model with independent calcium binding domian and temperature sensor.

**Figure supplement 4.** Renormalization of state-dependent interaction parameters.

low temperature is even larger. Thus our calculated change of $\Delta\chi$ from these experimental values is likely to be an underestimate. The change in $\Delta\chi$ indicates that while the overall allosteric coupling is favorable at both temperatures (i.e. the calcium-binding sites and the pore prefer to exist in 'like' conformations), the coupling interaction is much weaker at 37°C with respect to 21°C, which is also consistent with the relatively shallower dose response curve observed at 37°C.

In the context of a general allosteric model, let us consider the higher and lower asymptotes, $\chi_-$ and $\chi_+$, which are:

$$\chi_- = \ln L_0 \sum \frac{\theta_{UO}}{\theta_{UC}} \qquad (3a)$$

$$\chi_+ = \ln L_0 \sum \frac{\theta_{BO}}{\theta_{BC}} \qquad (3b)$$

where $L_0$ is the intrinsic equilibrium constant of pore opening. Note in the above equation $L_0$ term occurs in both expressions defining $\chi_-$ and $\chi_+$. Therefore, if the temperature-dependence of $L_0$ term was responsible for thermosensitivity of MthK channels, then both $\chi_+$ and $\chi_-$ would be equally affected. Also note that ligand association constant term, $K_{Ca}$ does not occur at all in the expressions describing two asymptotes. Thus, $\chi_-$ and $\chi_+$ are not going to be altered by temperature if the primary effect of temperature is to alter the ligand binding. As outlined previously (*Sigg, 2013*; *Chowdhury and Chanda, 2010*), this type of linkage analysis enables us to deconstruct the effect of mutations, drugs or other perturbations on any allosteric system in terms of their effects on various parameters.

The Hill transform of calcium dependence of $P_O$ values reveal that the $\chi_-$ is much more sensitive to temperature than $\chi_+$ and, therefore, one or both of the interaction terms, $\theta_{UO}$ and $\theta_{UC}$, must contribute to this higher temperature dependence of $\chi_-$. Although the analysis here is shown for a single ligand binding site, these conclusions are valid even for multiple ligand-binding sites (see 'Linkage analysis of Generalized Allosteric models' in Materials and methods section).

To better illustrate that our experimental conclusions follow the predictions of our proposed model of MthK gating, we simulated the Hill-plots at different temperatures allowing only one parameter of the model to be temperature-dependent at a time (*Figure 5B* and *Figure 5—figure supplement 1*). The simulations clearly show that temperature-dependent $\theta_{UO}$ or $\theta_{UC}$ is able to recapitulate the characteristic behavior of our experimental Hill-plots, namely a temperature-dependent $\Delta\chi$ and $\chi_-$ with a relatively modest temperature-sensitivity of $\chi_+$. Both $\theta_{UO}$ and $\theta_{UC}$ describes the interaction between the apo-sensor and the pore domain. Based on open-dwell time and Po measurements at low-calcium concentrations, we have previously argued that temperature causes a profound increase in the forward rate of channel opening (see *Figure 2*), which is also governed by $\theta_{UC}$. Therefore, we must conclude that the primary mechanism of temperature regulation involves decreased interaction energy between the closed pore and apo calcium-binding sites at elevated temperatures. We can draw a similar conclusion by comparing the Hill coefficient of the dose response curves at two temperatures. Under certain conditions, the Hill slope of a response curve is a measure of cooperativity (*Yifrach, 2004*) and the fact that the calcium dose-response curve at 37°C has a lower Hill coefficient than those obtained at 21°C further supports the notion that coupling interaction between various elements is reduced at higher temperatures in MthK.

## Discussion

Temperature modulates the functional activity of virtually all known proteins to a varying extent by altering the fluctuations of atoms, which frequently manifests itself as a change in the kinetics of a reaction. However, in some instances, it also results in altered equilibrium responses, as observed in temperature-sensitive ion channels. Mammalian thermoTRPs are founding members of temperature-sensitive ion channels that respond exquisitely to thermal stimuli (*Caterina et al., 1997*; *McKemy et al., 2002*; *Peier et al., 2002*). But in addition to thermoTRPs, many other structurally dissimilar ion channels such as STIM1-Orai complex are also activated by temperature (*Xiao et al., 2011*). This lack of structurally conserved temperature-sensing domain raises profound questions about the mechanisms of thermal sensing in biology.

Studies by several groups have led to identification of structural motifs that act as temperature-sensing domains whose activation is allosterically coupled to pore gating, analogous to regulation of ligand-dependent activation (*Arrigoni et al., 2016*; *Brauchi et al., 2006*; *Cordero-Morales et al., 2011b*; *Lishko et al., 2007*; *Takeshita et al., 2014*; *Fujiwara et al., 2012*; *Voets et al., 2007*). Here, we have examined the mechanism of temperature-sensitivity of archaebacterial MthK, which is an exemplar for studying calcium activation mechanisms. Single channel inside out patch clamp recordings of purified reconstituted MthK show that they are extremely sensitive to temperature between 20°C and 40°C. Over this range, the open probability of MthK IR increases about two orders of magnitude in low-calcium concentrations making it exquisitely sensitive to temperature, much like the prototypical eukaryotic thermosensors.

Studies on the TRPV1 which has been widely used as a model system to understand temperature-sensitivity shows that the region near the selectivity filter in the pore domain contributes to temperature-sensitivity (*Zhang et al., 2018*; *Kim et al., 2013*). Structural studies on TRPV3 channel, another thermosensitive ion channel, suggest that the region near the pore gates is responsible for temperature dependence (*Singh et al., 2019*). Here, we can rule out the possibility that the pore domain is primarily responsible for the observed temperature-dependence for the following reasons. First, linkage analysis predicts (See *Equations 3a and 3b*) that if the pore gating is intrinsically sensitive to temperature then the $ln(P_O/P_C)$ values will be equally displaced in both high and low asymptotes ($\chi_-$ and $\chi_+$). Our data clearly shows that at saturating calcium concentrations, the $ln(P_O/P_C)$ values are hardly affected by temperature in contrast to those at low calcium concentrations. Second, our single channel data of the MthK PO construct shows that increasing the temperature actually decreases the Po of the channel at high temperature by four-fold. The dependence is opposite to what one would expect if the temperature-dependence of the pore domain contributes to the observed thermosensitivity of MthK. Third, complementation plate assays at 28°C and 24°C clearly shows that the IPTG induction of the MthK ΔC construct rescues the growth of the LB2003 in low potassium (*Figure 1D*). Therefore, even though the MthK ΔC construct is not as effective as other two constructs, our data would argue that they form viable channels. While there is the theoretical possibility that intrinsic thermosensitivity of MthK PO construct is shifted outside our experimental range, the notion that the pore gating accounts for native temperature-dependence of MthK is incompatible with observed lack of thermosensitivity at high calcium concentrations.

Can temperature-sensitive calcium binding to MthK account for its thermal response? From *Equations 3a and 3b* we also note that both $\chi_-$ and $\chi_+$ values (and thus $\Delta\chi$) are independent of calcium binding affinity. Our simulations of a model where calcium binding affinity is the sole temperature dependent parameter of the system reinforces this point. The Hill plots at different temperatures have identical $\chi$ values (*Figure 5—figure supplement 1B*). We also envisioned a more complex scenario where a pore interacts with multiple calcium binding sites, of which some are temperature independent and some, temperature dependent (*Figure 5—figure supplement 2*). Simulated Hill plots of such a system show that the $\chi$ values will be invariant across different temperatures, irrespective of whether the calcium sensors interact directly with each other or not. Furthermore, if temperature dependent calcium binding affinity governs MthK's temperature sensitivity, we would expect the EC$_{50}$ of the dose response curves to change with temperature which is not observed in our data. Based on these lines of evidence, we can conclude that the temperature-dependent calcium binding alone cannot explain our functional observations. Nonetheless, it will be interesting to independently determine whether the calcium binding to the RCK domains of MthK is temperature-dependent.

It is important to note that our analysis does not necessarily rule out the possibility of existence of a discrete allosteric temperature sensing domain. However, unlike in classical models, this temperature sensing domain must be coupled not just to the pore domain but also to the RCK domain. If either of these interactions are missing, the limiting asymptotes $\chi_\pm$ will not exhibit characteristic temperature dependence observed here (*Figure 5—figure supplement 3*) (see also "Linkage analysis of Generalized Allosteric models" in Materials and methods section). Even within this nested allosteric framework, we should note that the effect of temperature can be simply distilled down to modulation of coupling between the RCK and pore domains. Whether this modulation is mediated directly by disrupting the interactions between RCK and pore or by a discrete temperature sensor which regulates RCK-pore coupling remains an open question. While further studies are required to delineate the structural and energetic mechanisms that underlie temperature-dependent gating in MthK, our findings highlight a novel mechanism of temperature-dependent gating in this ancient ion channel.

Interestingly, based solely on theoretical analysis of different allosteric models, *Jara-Oseguera and Islas, 2013* have previously proposed that temperature-sensitive allosteric coupling between stimulus sensing elements (such as voltage or ligand sensors) and a channel pore could underlie steep temperature-dependence of thermoTRPs. To the best of our knowledge, this study is the first known instance where temperature is shown to affect allosteric coupling rather than intrinsic equilibrium constant of a sensor domain.

In contrast to our proposed mechanism, several studies have suggested that temperature alters the equilibrium between two-gating states of a thermosensing domain (*Brauchi et al., 2006*;

*Raddatz et al., 2014*; *Cui et al., 2012*; *Cordero-Morales et al., 2011b*; *Jara-Oseguera and Islas, 2013*; *Yao et al., 2010*), which is coupled to the channel pore. This mechanism of allosteric regulation is equivalent to the classical allosteric models, which envision a specialized sensor involved in sensing the physical or chemical stimuli (*Figure 6*). The most notable example of this type of regulation is via the coiled-coil domains, which are found in many ion channels. Minor and colleagues have shown that the temperature-stability of C-terminal coiled-coil motif in the bacterial voltage-gated Na$^+$ channel directly regulates the opening and closing of the pore domain (*Arrigoni et al., 2016*). Similar studies on Hv1, a voltage-gated proton channel, shows that the coiled-coil motif in these channels also regulates its temperature-dependent activity (*Takeshita et al., 2014*; *Fujiwara et al., 2012*). Nevertheless, it should be pointed out that the temperature dependence of these channels is much lower than that of a prototypical temperature-sensitive TRP channel (*Clapham and Miller, 2011*) and for that matter, MthK as shown here.

Given the current known structures of MthK, we formed a speculation on how the coupling interactions between the pore and RCK domain is altered in a temperature-dependent manner. Recent X-ray and cryoEM structures of the full-length MthK show that the pore and RCK domains are not in physical contact except through the connecting linker (*Fan et al., 2020*; *Kopec et al., 2019*). This linker appears to be more ordered in the calcium free state than in the presence of calcium (*Fan et al., 2020*). The structure of closed MthK channel shows that the linker latches onto the RCK domains via hydrophobic and possible salt bridge interactions. Upon calcium binding, these hydrophobic interactions may be disrupted causing the linker to become disordered. Similarly, elevated temperature also may disrupt these interactions in the apo state and thereby manipulate the pore opening (*Fan et al., 2020*). Thus, one would posit that the linker may play a key role in coupling the apo-RCK domain to the conformation of the pore domain. Future studies probing the role of this linker helix may help provide a better understanding of the structural mechanisms that underlie temperature-dependent gating in these ion channels.

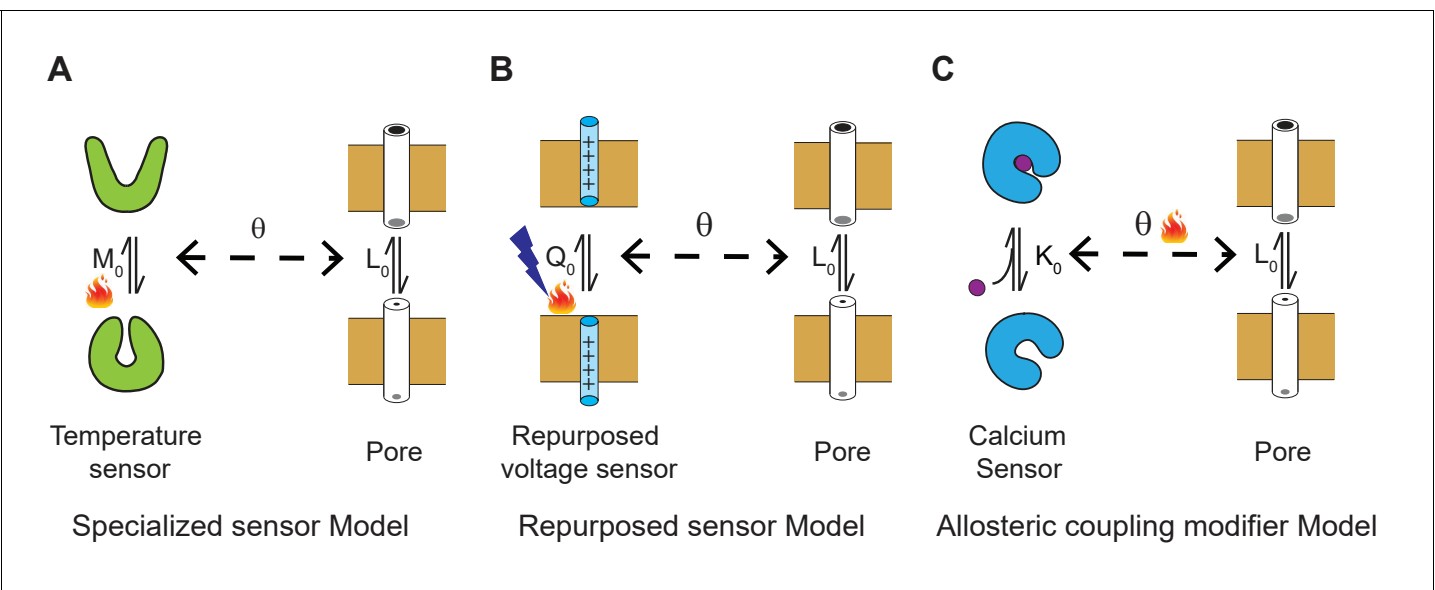

**Figure 6.** Three general allosteric mechanisms of temperature regulation. (A) Specialized sensor model. A classical allosteric model where the pore domain is allosterically coupled to a specialized domain whose conformation is temperature-dependent. $M_0$ defines the intrinsic temperature-dependent equilibrium constant for activation of a dedicated thermosensing domain. $L_0$ is the intrinsic equilibrium for pore opening and $\theta$ is the coupling interaction between the thermosensor and pore. Examples of specialized temperature sensors include bacterial sodium channels and Hv1 channels. (B) Polymodal sensor model. A variant of the classical allosteric model wherein the ligand or voltage-sensing domain is also sensitive to temperature. In this example, $Q_0$ is the voltage-dependent equilibrium constant whose voltage-dependent activation is also sensitive to temperature. In the engineered temperature-sensitive Shaker mutants, the mutations likely alter the temperature-dependence of voltage-sensor movement. (C) Allosteric modifier model. Here, $K_0$ defines the calcium binding affinity of the calcium sensor and the primary effect of temperature is to alter the allosteric coupling interaction between the ligand-sensing domain and pore domain.

# Materials and methods

**Key resources table**

| Reagent type (species) or resource | Designation | Source or reference | Identifiers | Additional information |
|---|---|---|---|---|
| Strain, strain background (*Escherichia coli*) | XL1-Blue | Agilent | 200236 | |
| Strain, strain background (*Escherichia coli*) | LB2003 | *Parfenova et al., 2006* This paper | | From Dr. I. Hänelt lab |
| Strain, strain background (*Escherichia coli*) | OverExpress C43(DE3) | Lucigen | 60446–1 | |
| Recombinant DNA construct | MthK FL (*M. thermoautotrophicum*) | This paper | UniProtKB-O27564 | From Dr. B. S. Rothberg lab |
| Recombinant DNA construct | MthK IR construct | This paper | | Delete the DNA region corresponds N-terminal 2–17 amino acid in MthK full length |
| Recombinant DNA construct | MthK ΔC construct | This paper | | A stop codon was introduced after H117 in MthK IR background |
| Peptide, recombinant protein | Thrombin | GE Healthcare | 27084601 | |
| Peptide, recombinant protein | Trypsin type I | Sigma-Aldrich | T8003 | |
| Peptide, recombinant protein | Trypsin inhibitor type II-O | Sigma-Aldrich | T9253 | |
| Peptide, recombinant protein | Ready-Lyse Lysozyme solutions | Lucigen | R1804M | |
| Peptide, recombinant protein | OmniCleave Endonuclease | Lucigen | OC7850K | |
| Chemical compound, detergent | n-Decyl-β-D-maltopyranoside (DM) | Anatrace | D322 | |
| Chemical compound, detergent | n-Dodecyl β-D-maltoside (DDM) | Anatrace | D310 | |
| Chemical compound, lipids | Soybean Polar Lipid Extract | Avanti | 541602C | |

## Cloning, expression, and purification of MthK in *E. coli*

The MthK K$^+$ channel gene from *M. thermoautotrophicum* was cloned into expression vector pQE82L as described previously (*Parfenova et al., 2006*). N-terminal (2–17 amino acid position) was deleted in MthK IR construct. MthK ΔC was generated by introducing a stop codon after H117 position in the MthK IR background. The constructs were expressed in *E. coli* XL1-Blue cell cultures and induced overnight at 24˚C with 1 mM isopropyl-β-D-thiogalactopyranoside (IPTG). Protein was purified according to the protocol described by *Jiang et al., 2002* with minor modifications. Cell pellets were broken by sonication and solubilized with 40 mM n-decyl-β-D-maltopyranoside (DM) (Anatrace) in lysis buffer (20 mM Tris, pH 8.0, 100 mM KCl and 20 mM imidazole). Insoluble fraction was pelleted by high-speed centrifugation at 20,500 rpm (50,228 g) for 40 min in SS-34 Rotor (Sorvall). The supernatant was loaded on either Talon Co$^{2+}$ affinity (Clontech) or Ni-NTA Agarose (Qiagen) columns. Nonspecific binding proteins were washed with lysis buffer containing 5 mM DM. To elute MthK, the imidazole concentration was increased to 250 mM. His tag was removed by Thrombin (GE) digestion (10 NIH units per 5 mgs of protein) for 2 hr at room temperature. The digested protein was concentrated and injected into Superdex-200 (10/300 GL) size exclusion column (GE) with 25 mM Hepes, pH 7.6, 100 mM KCl and 5 mM DM.

MthK pore only domain was purified as described previously (*Li et al., 2007*; *Posson et al., 2013*). Briefly, following gel filtration, the purified MthK IR was collected and incubated with 1:50 (w/w) Trypsin type I (Sigma, T8003) for 2 hr. The reaction was stopped by adding two-fold excess Trypsin inhibitor type II-O (Sigma, T9253) and the cleaved protein was purified on a Superdex-200 (10/300 GL) size exclusion column (GE) using buffer containing 25 mM Hepes, 100 mM KCl, 1 mM (n-Dodecyl β-D-maltoside) DDM, 5 mM DM pH 7.6 in 4˚C.

## Complementation assay

For plate complementation assay, LB2003 competent cells (*Parfenova et al., 2006*) were transformed and plated in Luria broth (LB) plates containing 100 mM KCl. pQE32 (Qiagen) plasmid was used as a negative control. After O/N incubation at 37°C, the colonies from each of the plates were pooled and after serial dilution, they were plated in low potassium (4 mM KCl) plates containing IPTG. Without IPTG plates were used as control. To normalize for differences in growth rate, all the plates were grown until MthK FL colonies were observed at the lowest dilution and then imaged (ChemiDoc MP imaging system, BIO-RAD).

For complementation assay in suspension, we used the high-potassium and low-potassium medium as described previously (*Parfenova and Rothberg, 2006*). Single colony of transformed LB2003 was picked from the overnight high $K^+$ plates (10 g Tryptone, 5 g Yeast extract, 10 g KCl, 10 g Agar and add MilliQ water to 1L) and then further grown overnight in high $K^+$ medium (10 g Tryptone, 5 g Yeast extract, 10 g KCl and add MilliQ water to 1L). Next day, the culture was diluted to $OD_{600}$ = 0.8 with low-$K^+$ medium (10 g Tryptone, 5 g Yeast extract, 10 g NaCl and add MilliQ water to 1L) and the channel expression was induced at 37°C with 1 mM IPTG for 3 hr. This induced culture was further diluted to $OD_{600}$ = 0.1 with low $K^+$ medium and incubated overnight at either high (37°C) or low (24°C) temperatures. 5 mM $BaCl_2$ was used as blocker. Independent replicates were obtained from starter cultures derived from other colonies on the same plate.

## Giant *E. coli* spheroplast

*E. coli* OverExpress C43 (DE3) competent cells (Lucigen) transformed with MthK IR were grown to an optical density of 0.7 for making giant spheroplasts. 6 ml of this liquid culture was combined with 55 ml LB containing antibiotic and 50 µg/ml cephalexin at 42°C. After 2 hr of incubation, 1 mM IPTG was added to this culture and incubated for another hour at 37°C. Following induction, the cells were harvested by low-speed centrifugation, the pellet was gently resuspended in 3 ml of 1 M sucrose. The resuspended cells were incubated with 240 µl Tris, pH 8.0, 67,500 units of Ready-Lyse Lysozyme solutions (Lucigen), 1600 units OmniCleave Endonuclease (Lucigen) and 20 µl of 500 mM EDTA. After incubation at room temperature for approximately 10 min, the digestion was stopped by adding 1 ml of stop buffer (0.8 M sucrose, 80 mM Tris, pH 8.0, 12 mM $MgCl_2$). The cells were aliquoted into PCR tubes, flash frozen in liquid nitrogen and stored in −80°C freezer.

## Reconstitution of the purified protein in liposome

This protocol is based on methods used to reconstitute KcsA in soybean polar lipids (*Chakrapani et al., 2007*). A total of 25 mg/ml lipids (Avanti) in chloroform were dried under argon and kept overnight in vacuum. The dried lipids were resuspended using bath sonication in 250 mM KCl, 30 mM Hepes, pH 7.6, and 0.1 mM $CaCl_2$ to a final concentration of 15 mg/ml. To the suspension, 5 mM DM was added so that the protein:lipid molar ratio was either 1:500 or 1:1000 for single-channel recording studies. Detergents were removed by O/N dialysis using either 25 KD cutoff Slide-A-Lyzer Dialysis cassette for MthK IR or 7 KD cutoff cassette for MthK pore. The dialysis buffer was refreshed next day, and after 4 hr, the proteoliposomes were aliquoted and stored at −80°C.

## Electrophysiology

For inside-out patch clamping of spheroplasts, the recording pipette was filled with 150 mM KCl, 20 mM $MgCl_2$, 15 mM Tris, pH 8.0, 0.1 mM $CaCl_2$ and 450 mM sucrose. The bath solution was 150 mM KCl, 20 mM $MgCl_2$, 15 mM Tris, pH 8.0 with varied calcium concentrations indicated in the main text. Sucrose was added in the bath solution to make up for differences in osmolarity. MthK currents were blocked using 0.3 mM $BaCl_2$. All the microelectrodes (Drummond) for electrophysiological measurements had a resistance of around 3.5 MΩ, and the tip size was controlled ~5.0–5.5 bubble number (*Mittman et al., 1987*). Digidata 1440A interface (Axon instrument) was used to collect the data with 250 kHz sampling rate and low-pass filtered at 5 kHz. Currents were elicited from a holding potential of −10 mV by a 300 ms pulse ranging from −100 mV to +100 mV stepping at 20 mV intervals.

Single-channel recordings were obtained by patch-clamping reconstituted proteoliposomes. 30 µl of proteoliposomes were placed on a clean glass slide and dried in a desiccator under vacuum at 4°C. The sample was then rehydrated with 50 µl buffer (250 mM KCl, 30 mM Hepes, pH 7.6, and 0.1

mM $CaCl_2$) for more than 2 hr, which yielded giant multilamellar vesicles (GMV). Symmetrical solution were used for the pipette solution and bath solution. The 0.1 mM $Ca^{2+}$ buffer contained 200 mM KCl, 30 mM Hepes, pH 7.6, 0.1 mM $CaCl_2$, and 30 mM sucrose. For other $Ca^{2+}$ buffers, they contained 200 mM KCl, 30 mM Hepes, pH 7.6. The calcium concentration was varied from 0.2 to 10 mM and the osmolality was adjusted with sucrose. For MthK PO, symmetrical buffers containing 200 mM KCl, 30 mM Hepes, 1 mM $CaCl_2$, 27 mM sucrose, pH 7.6 were used. For temperature control, SC-20 dual in-line heater/cooler (Warner Instrument) was used with single-channel bipolar temperature controller CL-100 (Warner Instrument). Recording temperatures were monitored with a bead thermistor (TA-29; Warner Instrument) within 5 mm of the microelectrode tip. All the traces were recorded at −100 mV with 25 kHz sampling rate and low-pass filtered at 2–5 kHz.

## Data analysis

All statistical tests were performed using OriginPro program. Data are presented as mean ± SEM. Student's two sample T-test was used to evaluate the statistical significance of the results of two independently collected pools of data assuming non-equal variance. $p > 0.05$ was considered statistically non-significant; *, $p \leq 0.05$; **, $p \leq 0.01$.

For macroscopic recordings, currents were normalized to the maximum currents at 42°C in presence of calcium. Since MthK channels exhibit voltage-dependent rectification at positive potentials (*Li et al., 2007*), steady-state currents at −100 mV were used for analysis of macroscopic conductance changes. Single-channel data was digitally low pass filtered at 1.4 kHz and analyzed using Clampfit 9.0 (Axon Instrument). The integrated Single-channel Search module was used to detect any open event that is longer than 0.04 ms. For the coupling energy analysis, data with only 1°C variance from the indicated temperature was selected.

$nP_O$ is defined as

$$nP_O = \sum_{1}^{n} iP_i \tag{4}$$

where $n$ is the number of channels in a patch, $i$ indicates the number of the open channel and $P_i$ is the probability of the corresponding level. For every condition, we determined the total number of channels in a patch by going to high Po values either by heat or high calcium. We cannot use this method for MthK PO which do not respond to calcium. In this case, we did not calculate the absolute *Po* but only relative changes in *Po*.

The temperature coefficient ($Q_{10}$) was calculated using following equation:

$$Q_{10} = \left(\frac{A_2}{A_1}\right)^{\frac{10}{T_2 - T_1}}, \tag{5}$$

where $A_2$ and $A_1$ are *Po* at the two different temperatures.

For exponential fitting of open-dwell time, we used Clampfit program. The open-dwell time histogram was replotted by square root of y axis and logarithmic treatment of the x axis. Then the plot was fitted with Exponential, log probability equation:

$$f(t) = \sum_{i=1}^{n} P_i e^{[\ln(t) - \ln(\tau_i)]} e^{\ln(t) - \ln(\tau_i)}, \tag{6}$$

where t is the independent variable (dwell time), $P_i$ is the percentage of the term i, and $\tau_i$ is the time constant of term i. We used maximum likelihood as the minimization method with maximum iterations as 5000. Different term of models is compared automatically with confidence level set as 0.95.

For coupling energy analysis, the open probability (*Po*) was fitted to a Hill plot using the following equation:

$$P_O = A + (B - A) * \frac{x^n}{K^n + x^n}, \tag{7}$$

where *A* is the value at low calcium, *B* is the plateau value at saturation calcium, *K* is equal to $K_d$ of

the calcium-binding, and $n$ is the Hill coefficient, the fitted figures were plotted in *Figure 4—figure supplement 1*. With these constants, we can fit ln [*Po/(1-Po)*], which was shown in *Figure 4C*.

## Linkage analysis of generalized allosteric models

### Equivalence between classical allosteric model with a single coupling parameter and an allosteric model with state-dependent interaction parameters

We consider the binary elements model depicted in *Figure 5A*, whose parameters are defined in the main text. Let us consider the reference energy level for this system to be the closed pore-apo calcium sensor. The partition function can be written as:

$$Z = \theta_{UC} + L_0\theta_{UO} + K_{ca}x\theta_{BC} + L_0K_{ca}x\theta_{BO}, \tag{8}$$

where $x$ is the calcium concentration. Now we can divide *Equation 8* with $\theta_{UC}$ to get:

$$\frac{Z}{\theta_{UC}} = 1 + \frac{L_0\theta_{UO}}{\theta_{UC}} + \frac{K_{ca}x\theta_{BC}}{\theta_{UC}} + \frac{L_0K_{ca}x\theta_{BO}}{\theta_{UC}}, \tag{9}$$

We define, $L_0' = \frac{L_0\theta_{UO}}{\theta_{UC}}$ and $K_{Ca}' = \frac{K_{ca}\theta_{BC}}{\theta_{UC}}$ and use these parameters in *Equation 9* to get:

$$\frac{Z}{\theta_{UC}} = 1 + L_0' + K_{Ca}'x + L_0'K_{Ca}'x\frac{\theta_{UC}\theta_{BO}}{\theta_{UO}\theta_{BC}}, \tag{10}$$

Next, we define $\theta = \frac{\theta_{UC}\theta_{BO}}{\theta_{UO}\theta_{BC}}$, which is simply the ratio of 'like-state' interactions versus 'unlike-state' interactions and is a measure of cooperativity between the two binary elements. Using $\theta$ and redefining $\theta_{UC}$ as the reference energy level, *Equation 10* becomes:

$$Z' = 1 + L_0' + K_{Ca}'x + L_0'K_{Ca}'x\theta, \tag{11}$$

The above equation (*Equation 11*) is mathematically analogous to the partition function of a binary elements model where the coupling interaction between the binary elements is restricted to the open-pore and bound calcium sensor as would be the case in classical four state allosteric models (*Figure 5—figure supplement 4*). Yet, by introducing parameter normalizations (to get to *Equation 10* from 9) and redefining the reference energy level, we are able to arrive at a similar mathematical expression even for a system where there are multiple conformational-state dependent interactions between the binary elements. However, it is important to realize that in this normalized form, the contributions of the various state-dependent interactions are incorporated within each of the 'normalized' parameters.

### Analytical derivation of temperature dependence of χ-values for a model with a discrete temperature sensor

A more conventional model often used to describe temperature-dependent gating of ion channels invokes a specific temperature-sensing domain, allosterically coupled to the channel pore (*Diaz-Franulic et al., 2016*). As shown in *Figure 5—figure supplement 3A*, this allosteric domain is associated with a temperature-dependent activation constant, $M_0$, and its interactions with the pore and ligand-binding domains are associated with allosteric factors C and E, respectively. In this model, the coupling between the pore and ligand-binding domain is represented by the single allosteric parameter, D. For such a model, we are interested in understanding how the χ-values would change with temperature. The χ-values for ligand driven transformation of the channel can be represented as:

$$\chi_- = \ln\frac{L_0(1+M_0C)}{1+M_0}, \tag{12}$$

$$\chi_+ = \ln\frac{L_0D(1+M_0CE)}{1+M_0E}, \tag{13}$$

$$\Delta\chi = \ln\frac{D(1+M_0CE)(1+M_0)}{(1+M_0C)(1+M_0E)}, \tag{14}$$

We are interested in deriving the expressions for the temperature dependence of the χ-values which arises due to the logarithmic terms in *Equations 12–14*. Therefore, we differentiate each of the above equations with respect to temperature (T), noting that of all the different parameters only $M_0$ is temperature-dependent. The following equations are the temperature differentials of the χ-values:

$$\frac{\partial\chi_-}{\partial T} = \frac{\partial M_0}{\partial T}\frac{C-1}{(1+M_0)(1+M_0C)}, \tag{15}$$

$$\frac{\partial\chi_+}{\partial T} = \frac{\partial M_0}{\partial T}\frac{E(C-1)}{(1+M_0E)(1+M_0CE)}, \tag{16}$$

$$\frac{\partial\Delta\chi}{\partial T} = \frac{\partial M_0}{\partial T}\frac{(E-1)(C-1)\left(1-M_0^2CE\right)}{(1+M_0)(1+M_0C)(1+M_0E)(1+M_0CE)}, \tag{17}$$

From Eqs.15-17, we can see that when C = 1, all three differentials are zero indicating their lack of temperature dependence. When C ≠ 1 but E = 1, both $\chi_+$ and $\chi_-$ are temperature-dependent but $\Delta\chi$ is temperature independent. Only when both E and C are different from unity, are the differentials non-zero and unequal (*Figure 5—figure supplement 3B* shows model simulations).

Another interesting observation is that if $M_0$ is a monotonically increasing function of temperature, then both $\chi_-$ and $\chi_+$ are monotonically temperature-dependent. For instance, in the case where C > 1 and $\frac{\partial M_0}{\partial T}$ >0, the sign of the differential is positive and thus as temperature is increased both $\chi_-$ and $\chi_+$ will keep increasing. However, $\Delta\chi$ may be a non-monotonic function of temperature. For a given value of E and C and a fixed sign of the differential $\frac{\partial M_0}{\partial T}$, the sign of the temperature differential of $\Delta\chi$ will change depending on whether $M_0$ is greater or less than $1/\sqrt{CE}$. Since $M_0$ is itself temperature-dependent, it is possible that over a specific temperature range $M_0 < 1/\sqrt{CE}$ while at a different temperature regime $M_0 > 1/\sqrt{CE}$. However, this crossover temperature might not be accessible to electrophysiological recordings.

## Deconstruction of the coupling energy for a Ligand gated ion channel with multiple binding sites (General model with N binding sites)

Let us consider a ligand gated ion channel, comprising a pore that can exist in two conformations, open (O) and closed (C), and 'N' different ligand-binding sites, each with an intrinsic ligand-binding affinity, $K_i$. Each ligand-binding site, in its apo or liganded states, interacts with the pore in its open or closed conformations. These interactions cumulatively influence the conformational bias of the pore and regulate ligand-induced channel gating. Furthermore, the ligand-binding sites can interact with each other and these interactions influence the cooperativity of ligand binding independent of the conformational state of the pore. For this system, we define, $\theta_{UC}^{(i)}$ and $\theta_{BC}^{(i)}$ as the coupling constants representing the interaction energies between the closed pore and the unbound and ligand-bound states of the $i^{\text{th}}$ ligand-binding site (indicated by the superscripts (*i*)), respectively. Similarly, $\theta_{UO}^{(i)}$ and $\theta_{BO}^{(i)}$ are the coupling constants representing the interaction energies between the open pore and the unbound and ligand-bound states of the $i^{\text{th}}$ ligand-binding site, respectively. The coupling constants representing the direct interactions between the ligand-binding sites are $\alpha_{iU,jU}$, $\alpha_{iB,jU}$, $\alpha_{iU,jB}$ and $\alpha_{iB,jB}$, where the subscripts indicate whether the $i^{\text{th}}$ (i = 1, 2, ... N) and $j^{\text{th}}$ (j = 1, 2, ... N) ligand-binding sites are in their bound (B) or unbound states (U).

Under an allosteric framework, this ligand gated ion channel is capable of existing in $2^{N+1}$ conformations, with $2^N$ closed and $2^N$ open states. We define the 'contracted partition of the closed states', $Z_C$, for this system as:

$$Z_C = \sum_{l=0}^{N}B_l^C x^l, \tag{18}$$

where x represents the ligand concentration, $B_l^C$ is the 'Boltzmann weight' of the $l^{th}$ liganded closed state (which is informed by the ligand-binding affinities, the interactions of the ligand-binding sites with the pore and between each other) and the summation is over the number of ligand-binding sites. This (contracted) partition function is analogous to the 'binding polynomial' used in Wyman's linkage theory (58). For instance, for the unliganded state, $B_0^C = \prod_{i=1}^N \theta_{UC}^{(i)} \left( \prod_{i=1}^N \prod_{j=1, j\neq i}^N \alpha_{iU,jU} \right)$ since the energy of the state is informed by the interactions of all the apo ligand-binding sites with the closed pore and between each other. Similarly for the fully liganded state, $B_N^C = \prod_{i=1}^N K_i \prod_{i=1}^N \theta_{BC}^{(i)} \left( \prod_{i=1}^N \prod_{j=1, j\neq i}^N \alpha_{iB,jB} \right)$ since the energy of this state is informed by the interaction energy of the ligands with the ligand-binding sites (which govern $K_i$) and the interactions of holo ligand-binding sites with the closed pore and between each other. The Boltzmann weights for the states where only a few ligand-binding sites can be deduced accordingly using a combination of the coupling constants described.

The 'contracted partition function of the open states', $Z_O$, for this system can be defined as:

$$Z_O = \sum_{l=0}^N B_l^O x^l, \tag{19}$$

where, $B_l^O$ is the 'Boltzmann weight' of the $l^{th}$ liganded open state and the summation is over the number of ligand-binding sites. In addition to the ligand-binding affinities and coupling constants, $B_l^O$, is also determined by the intrinsic pore opening equilibrium constant, $L_0$, which governs the innate energetic bias of the open conformation of the pore relative to its closed conformation. As examples, for the unliganded state, $B_0^O = L_0 \prod_{i=1}^N \theta_{UO}^{(i)} \left( \prod_{i=1}^N \prod_{j=1, j\neq i}^N \alpha_{iU,jU} \right)$ and for the fully liganded state, $B_N^O = L_0 \prod_{i=1}^N K_i \prod_{i=1}^N \theta_{BO}^{(i)} \left( \prod_{i=1}^N \prod_{j=1, j\neq i}^N \alpha_{iB,jB} \right)$.

With these definitions, the Hill-transformed open channel probability, $H_L\{P_O\}$, is defined as:

$$H_L\{P_O\} = \ln\left(\frac{P_O}{1-P_O}\right) = \ln\left(\frac{Z_O}{Z_C}\right) = \ln\left(\frac{\sum_{l=0}^N B_l^O x^l}{\sum_{l=0}^N B_l^C x^l}\right), \tag{20}$$

Note that the ratio $P_O/(1-P_O)$ represents a macroscopic equilibrium constant of channel opening which varies with ligand concentration due to the interaction between the channel pore and ligand-binding sites. $H_L\{P_O\}$ is a non-linear function of ligand concentration, x, and depends on several microscopic parameters of the system. However at limiting ligand concentrations it simplifies greatly. In the absence of ligand, or at concentrations where the ligand-binding sites remain unoccupied by the ligand, $H_L\{P_O\}$ reduces to:

$$\lim_{x\to 0} H_L\{P_O\} = \ln\left(\frac{B_0^O}{B_0^C}\right) = \ln\left(\frac{L_0 \prod_{i=1}^N \theta_{UO}^{(i)}}{\prod_{i=1}^N \theta_{UC}^{(i)}}\right) = \chi_-, \tag{21}$$

Thus at low ligand concentrations, $H_L\{P_O\}$ reaches a limiting value, $\chi_-$, which depends on the intrinsic pore opening constant and the interactions of the apo ligand-binding sites with the pore.

In a similar way, we can deduce that $H_L\{P_O\}$ reaches a limiting value, $\chi_+$, at saturating ligand concentrations. To this end, we must realize first that at very high concentrations, $B_N^O x^N \gg B_{N-1}^O x^{N-1} \gg \cdots \gg B_1^O x \gg\gg B_0^O$ and $B_N^C x^N \gg B_{N-1}^C x^{N-1} \gg \cdots \gg B_1^C x \gg B_0^C$, simply because $x \to \infty$. Thus, under these limiting conditions, $H_L\{P_O\}$ can be expressed as:

$$\lim_{x\to\infty} H_L\{P_O\} = \ln\left(\frac{B_N^O}{B_N^C}\right) = \ln\left(\frac{L_0 \prod_{i=1}^N \theta_{BO}^{(i)}}{\prod_{i=1}^N \theta_{BC}^{(i)}}\right) = \chi_+, \tag{22}$$

This indicates at very high ligand concentrations when the occupancy of fully liganded channel is far greater than the occupancies of the partially liganded state, $H_L\{P_O\}$ reaches another limiting value which depends on the intrinsic pore opening constant and the interactions of the holo ligand-binding sites with the pore. The difference between the two limiting values, $\chi$, is:

$$\Delta\chi = \chi_+ - \chi_- = \ln\left(\frac{\prod_{i=1}^N \theta_{BO}^{(i)}}{\prod_{i=1}^N \theta_{BC}^{(i)}} \cdot \frac{\prod_{i=1}^N \theta_{UC}^{(i)}}{\prod_{i=1}^N \theta_{UO}^{(i)}}\right) = \sum_{i=1}^N \ln\left(\frac{\theta_{BO}^{(i)} \theta_{UC}^{(i)}}{\theta_{BC}^{(i)} \theta_{UO}^{(i)}}\right), \tag{23}$$

$\Delta\chi$ is thus independent of the ligand-binding affinities, direct interactions of the ligand-binding sites between each other and the intrinsic pore opening equilibrium constants. Instead $\Delta\chi$ depends only on the direct interactions of the ligand-binding sites with the pore. Additionally, $\Delta\chi$ can be realized to be the difference between 'like-state' interaction energies and 'unlike-state' interaction energies between the pore and the ligand-binding sites that is a favorable interaction between the like-states (which implies $\theta_{BO}^{(i)} > 1$ or $\theta_{UC}^{(i)} > 1$) will lead to larger $\Delta\chi$ values while favorable interaction between unlike states (which implies $\theta_{BC}^{(i)} > 1$ or $\theta_{UO}^{(i)} > 1$) will lead to smaller $\Delta\chi$ values.

$\Delta\chi$, calculated as the difference between the limiting values of Hill-transformed, experimentally measured Po, allows us to extract the 'net coupling strength' between all the ligand-binding sites and the pore without fitting the measured experimental responses to predicted/simulated responses to candidate models.

# Acknowledgements

We thank Dr. B S Rothberg and Dr. I Hanelt for generously sharing MthK plasmid and LB2003 cells, respectively. We are also grateful to Dr. S Chakrapani for her advice on patch-clamping proteoliposomes, Dr. S Sukharev for help with recordings from bacterial spheroplasts, Drs. C Miller and N Last for teaching us to record using Orbit mini and, Drs. C Nimigean and D Posson for generously sharing their protocol for pore domain purification. Drs. K Henzler-Wildman, VA Klenchin and H Chen helped with various aspects of protein purification. We thank other members of the Chanda lab for their valuable feedback and comments. This work was supported by funding from the National institutes of Health R01NS081293 (BC), R35NS116850 (BC) and UW-Madison Biophysics training program.

# Additional information

### Competing interests
Baron Chanda: Reviewing editor, *eLife*. The other authors declare that no competing interests exist.

### Funding

| Funder | Grant reference number | Author |
| --- | --- | --- |
| National Institute of Neurological Disorders and Stroke | R01NS081293 | Baron Chanda |
| National Institute of Neurological Disorders and Stroke | R35NS116850 | Baron Chanda |

The funders had no role in study design, data collection and interpretation, or the decision to submit the work for publication.

### Author contributions
Yihao Jiang, Conceptualization, Data curation, Formal analysis, Investigation, Methodology, Writing - original draft, Writing - review and editing; Vinay Idikuda, Formal analysis, Validation, Investigation, Writing - review and editing; Sandipan Chowdhury, Software, Formal analysis, Investigation, Writing - original draft, Writing - review and editing; Baron Chanda, Conceptualization, Resources, Supervision, Funding acquisition, Visualization, Writing - original draft, Project administration, Writing - review and editing

### Author ORCIDs
Yihao Jiang (iD) https://orcid.org/0000-0001-7694-3089
Vinay Idikuda (iD) https://orcid.org/0000-0001-6324-5839

Sandipan Chowdhury [iD] https://orcid.org/0000-0002-0695-7968
Baron Chanda [iD] https://orcid.org/0000-0003-4954-7034

Decision letter and Author response
Decision letter https://doi.org/10.7554/eLife.59055.sa1
Author response https://doi.org/10.7554/eLife.59055.sa2

## Additional files

### Supplementary files

• Transparent reporting form

### Data availability

Source data for all the plots were put together in a single excel file.

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
