## [Decision Letter]

**Acceptance summary:**

This manuscript investigates the mechanisms of temperature activation in a prokaryotic calcium-gated potassium channel, MthK, which thrives at temperatures around 65C. This work is of interest in terms of MthK serving as a model for analysis of the mechanistic and structural bases of ion channel gating, as well as a model for understanding the physiology of thermophiles.

**Decision letter after peer review:**

Thank you for submitting your article "Activation of MthK is exquisitely regulated by temperature" for consideration by *eLife*. Your article has been reviewed by three peer reviewers, and the evaluation has been overseen by Merritt Maduke as the Reviewing Editor and Kenton Swartz as the Senior Editor. The reviewers have opted to remain anonymous.

The reviewers have discussed the reviews with one another, and the Reviewing Editor has drafted this decision to help you prepare a revised submission.

As the editors have judged that your manuscript is of interest, but as described below that additional experiments are required before it is published, we would like to draw your attention to changes in our revision policy that we have made in response to COVID-19 (https://elifesciences.org/articles/57162). First, because many researchers have temporarily lost access to the labs, we will give authors as much time as they need to submit revised manuscripts. We are also offering, if you choose, to post the manuscript to bioRxiv (if it is not already there) along with this decision letter and a formal designation that the manuscript is "in revision at *eLife*". Please let us know if you would like to pursue this option.

Summary:

The manuscript by Jiang et al. investigates the mechanisms of temperature activation in a prokaryotic calcium-gated potassium channel, MthK, which thrives at temperatures around 65{degree sign}C. This work is of interest in terms of MthK serving as a model for analysis of the mechanistic and structural bases of ion channel gating, as well as a model for understanding the physiology of thermophiles.

The authors perform different types of experiments to first show that temperature indeed activates the channel: functional complementation assays using MthK to rescue, macroscopic current recording with patch-clamp of MthK-expressing *E. coli* spheroplasts, single-channel recordings with patch clamp from vesicles containing purified MthK. Through detailed quantitative analysis of MthK gating as a function of temperature, Ca^2+^ concentration, and the presence of the RCK Ca^2+^-binding domain, they conclude that temperature affects the allosteric coupling mechanism between MthK's RCK and pore domains.

The data and analysis are of high quality, and the conclusions present a novel, intriguing way in which temperature modulates the activity of an ion channel. However, the reviewers have several major concerns that must be addressed.

Essential revisions:

1) It appears that the bacterial growth-rescue experiments (Figure 1) were only carried out once. This raises concerns about their reproducibility. Information on experiment replication should be included. It appears that the authors simply performed densitometric measurements based on imaging of the cultures, which is not a robust means of quantification. Ideally, quantification should be performed by measuring OD600 of mini suspension cultures in triplicate for each condition. Also, it would be ideal to include a control with a MthK channel blocker, as it is possible that some of the growth deficiencies in truncated constructs result from temperature-dependent misfolding or other alternative mechanisms unrelated to potassium permeation through the channels.

2) The authors should provide data showing that the steep temperature-dependent increase in Po that occurs between 32 and 37C is reversible, showing that once channels have reached maximal activation, Po returns back to the same level it had at the beginning of the experiment at room temperature. All analysis and mechanistic interpretations assume that everything happens at equilibrium, but no evidence is provided to support this key assumption. Given that temperature-activated TRP channels exhibit prominent hysteresis when activated by heating, it is also important to show that the same Po at 37C degrees can be consistently achieved in repeated stimulations in the same patch. If this were not the case, then the thermodynamic analysis performed on the data would not be valid. Notably, MthK was chosen as a model system because eukaryotic temperature-activated channels are polymodal and their responses to temperature often involve out-of-equilibrium processes. Yet, no data is provided showing that MthK – which is also a polymodal receptor – is exempt from these same issues.

3) In Figure 2 we only see the Po increases at 1 temperature value, 39 degrees, while for all other values tested, there is no effect, which is somewhat odd, and also at odds with the more monotonic increase in Po with temperature in Figure 1—figure supplement 1E. Is this real and is it happening in all experiments? The authors should discuss this.

4) The temperature-dependence of the macroscopic currents in Figure 1—figure supplement 1 appears negligible. Given the enormous fold-differences in rescue between room temperature and 36C in Figure 1, one would expect to see similar steeply temperature-dependent changes in the macroscopic currents if the rescue is indeed reflecting changes in potassium conductance, that should be evident even if incomplete Ba^2+^-block results in over-subtraction of currents at higher temperatures. Why were experiments not carried out at lower Ca^2+^-concentrations (or in its absence), where temperature-dependence might be more pronounced? The authors should also show recordings and corresponding group data for spheroplasts that are not expressing MthK channels at different temperatures and Ca^2+^-concentrations. The authors should discuss the possibility that temperature-dependence of MthK channels is dependent on the expression system and possibly the lipid environment. Is it possible that the difference in bacterial strains used in the rescue experiments vs the macroscopic current recordings could be responsible for the apparent lack of temperature sensitivity in the latter?

5) With the single-channel conductance in Figure 2, the authors show a graph (C) where conductance goes up with temperature, with a low value at 21 degrees, but this is not visible from traces in A, or from those in Figure 2—figure supplement 1. But these all look identical. This issue should be revisited.

6) A serious concern is the apparent lack of independent replicates in the single-channel data. For example, it is clearly stated that all traces in Figure 2A are from one patch but it is not stated how many patches were recorded, what is the average behavior of these individual recordings. There is no plot where a mean value with errors from recordings in different patches is shown. Has this been done for more than one patch? Graphs in C and D should show averages over multiple patches, but they do not. The open time plot for example is actually a mean open time from just one experiment with many events. For the bar graph in D, the authors say: "error bars are SEM calculated by bootstrapping from 10 sweeps of 30 s length". This is just an n of 1 and needs to be repeated.

We have the same concern for the data shown in Figures 3 and 4. In Figure 3, only one experiment is analyzed (performed?) for the pore-only MthK channel. In Figure 4, it appears that the two Po vs Ca^2+^ plots in C are derived from just one experiment at each Ca^2+^ concentration, where temperature was changed from 21 to 37C. Please clarify if this is an n of 1 for each calcium concentration, in which case it needs to be repeated. If not, the number of separate patches should be clearly stated in the figure legend.

For all of these experiments, the authors must clearly indicate the number of patches from which data was collected for each condition and channel type, clearly denoting which represent independent replicates, and if they are biological or technical replicates. It is essential that the main findings in Figure 3 and Figure 4C are supported by at least three independent replicates, ideally more.

7) For the NPo measurements in Figure 3A; based on the recordings shown, it is difficult to see how the data at 36C (red trace) have a 4-fold lower NPo than the data at 20C (blue trace), unless the red trace includes some very long inactive periods. The authors should explain this further.

8) It would be useful for the readers that the authors discuss their proposed mechanism of how temperature increases Po from the perspective of the existing structures of MthK, rather than only abstractly. Especially since the mechanism invoked by the authors involves linkage between the ligand-binding domain and the pore domain and these linkers are actually resolved in the closed channel structure. The authors should not use "helices" to describe the linker, since it is only a short loop that connects the end of the TM2 helix with the RCK domain, at least in the closed state structure, where it is visible.

[Editors' note: further revisions were suggested prior to acceptance, as described below.]

Thank you for submitting your article "Activation of an archaeal ion channel is exquisitely regulated by temperature" for consideration by *eLife*. Your article has been reviewed by the three original peer reviewers, and the evaluation has been overseen by Merritt Maduke as the Reviewing Editor and Kenton Swartz as the Senior Editor. The reviewers have opted to remain anonymous.

The reviewers have discussed the reviews with one another and the Reviewing Editor has drafted this decision to help you prepare a revised submission.

Summary:

The manuscript by Jiang et al. investigates the mechanisms of temperature activation in a prokaryotic calcium-gated potassium channel, MthK, which thrives at temperatures around 65C. This work is of interest in terms of MthK serving as a model for analysis of the mechanistic and structural bases of ion channel gating, as well as a model for understanding the physiology of thermophiles.

The authors perform different types of experiments to first show that temperature indeed activates the channel: functional complementation assays using MthK to rescue, macroscopic current recording with patch-clamp of MthK-expressing *E. coli* spheroplasts, single-channel recordings with patch clamp from vesicles containing purified MthK. Through detailed quantitative analysis of MthK gating as a function of temperature, Ca^2+^ concentration, and the presence of the RCK Ca^2+^-binding domain, they conclude that temperature affects the allosteric coupling mechanism between MthK's RCK and pore domains.

The data and analysis are of high quality, and the conclusions present a novel, intriguing way in which temperature modulates the activity of an ion channel. However, the reviewers have a few remaining concerns. These can be addressed by changes to the manuscript without any additional experiments.

1) The authors do not provide enough evidence to strongly conclude that the pore domain has no contribution to temperature sensing. First, in the complementation assays, there is no evidence that δ-C forms viable channels. These data should be removed. Second, there are dramatic temperature-dependent changes in the gating of single channels lacking the C-terminus – the mean closed dwell times are noticeably shorter (see Figure 3A and Figure 3—figure supplement 1C), which is what largely contributes to a temperature-dependent change in Po for the constructs with an intact C-terminus. It is thus undeniable that the construct without a C-terminus undergoes temperature-dependent conformational transitions – it might just be that these don't lead to an increase in Po due to the large disruption in the overall protein energy landscape caused by the deletion of a functionally important part of the channel. Finally, the range of high temperatures explored by the authors is narrow, particularly compared to the ranges over which heat-sensitive TRP channels are activated, which is above the 36 C (or 39 C, although these data were not included in the manuscript). Whereas it is true that this limitation applies to all other constructs, it is also true that all other constructs did show temperature-dependent changes in Po below 39 C. This point is central to the conclusions in the manuscript and must therefore be discussed more factually. The conclusions regarding the MthK-PO construct should be toned down throughout the manuscript.

2) The Hill plots in Figure 4C do not allow the authors to accurately determine the value of the asymptote at the lower Ca^2+^ concentrations (X-) and thus the value of deltaX is also equally undetermined; there is over an order of magnitude change in the value of the Hill plot from the lower Ca^2+^ concentration to the second lowest at both temperatures, arguing against a plateauing over the examined range of concentrations. Without this, no quantitative analysis of the data can be obtained. Yet, the data do show that qualitatively the lower asymptote, wherever it might fall, is more temperature-dependent than the upper asymptote, which still supports the authors conclusions. Therefore, the argument must be framed in a qualitative manner. (We note that you provide estimates of coupling energies in the context of "classical allosteric analysis", citing Horrigan and Aldrich, but we didn't see that type of analysis presented in this manuscript. We also note that your results are quantitatively different from those reported by Thomson and Rothberg, 2010. Framing your results in a qualitative rather than quantitative way will resolve these issues.

We suggest you could include one paragraph where you examine how the theoretical framework (please elaborate) applies specifically to your experimental data. You could test whether an *X_-_* asymptote can truly be inferred from the data (perhaps comparing between fits with different limiting values). We would like you to provide a compelling argument that no other model (for example, one with a temperature-dependent Ca^2+^-association constant – see Figure 5—figure supplement 1B) is consistent with your data. It would strengthen the paper if you can show that your Hill plot data at the two temperatures cannot be described by two curves that have the same *X_-_*asymptote at a much lower ln(Po/(1-Po)) value than those observed in the data.

Revisions expected in follow-up work:

Experiments essential to support the conclusion that the pore domain has no contribution to temperature sensing are as follows. First, the complementation assays should include western blots showing that the expression of the construct lacking the C-terminus is not heavily compromised in the context of this assay. Without these data, it is not possible to conclude that the lack of rescue by this construct is due to an absence of temperature-dependent activation, particularly because that construct also fails to rescue at room temperature. Second, a higher range of temperatures should be explored. In the TRP channel literature, it is standard to go at least over 40 C, because below those temperatures no steeply temperature dependent responses are usually observed; MthK comes from a thermophilic organism that grows optimally between 55-60C, so a lack of response at 37C is not sufficient to conclude there is no temperature response.

[Editors' note: further revisions were suggested prior to acceptance, as described below.]

Thank you for submitting your article "Activation of the archaeal ion channel MthK is exquisitely regulated by temperature" for consideration by *eLife*. Your article has been reviewed by one peer reviewer, and the evaluation has been overseen by a Reviewing Editor and Kenton Swartz as the Senior Editor. The reviewers have opted to remain anonymous.

The reviewers have discussed the reviews with one another and the Reviewing Editor has drafted this decision to help you prepare a revised submission.

Summary:

The manuscript by Jiang et al. investigates the mechanisms of temperature activation in a prokaryotic calcium-gated potassium channel, MthK, which thrives at temperatures around 65C. This work is of interest in terms of MthK serving as a model for analysis of the mechanistic and structural bases of ion channel gating, as well as a model for understanding the physiology of thermophiles.

The authors perform different types of experiments to first show that temperature indeed activates the channel: functional complementation assays using MthK to rescue, macroscopic current recording with patch-clamp of MthK-expressing *E. coli* spheroplasts, single-channel recordings with patch clamp from vesicles containing purified MthK. They perform a detailed quantitative analysis of MthK gating as a function of temperature, Ca^2+^ concentration, and the presence of the RCK Ca^2+^-binding domain, and they suggest that temperature may modulate the allosteric coupling mechanism between MthK's RCK and pore domains.

The data and analysis are of high quality, and the authors suggest a novel, intriguing way in which temperature modulates the activity of an ion channel. However, there are limitations in some of the measurements that preclude definitive support of the conclusions as written. We therefore request that the authors develop their arguments more carefully and soften the conclusions.

Revisions:

The model simulations in Figure 5 and Figure 5—figure supplements 1-3 are adequate for a review article on linkage analysis, but are not useful for analyzing the experimental data that the paper is about; they are all graphed at distinct scales than the data in Figure 4C, and it is impossible to determine which of those model predictions actually describes the data accurately and which ones don't.

Using the data provided by the authors (Figure 4—figure supplement 1), one of the reviewers fit the Po vs [Ca^2+^] relations with the Hill equation used by the authors without applying any constraints on the fitting (red curves in the graph) and by constraining fits at both 21 and 37C to have the same *X_-_*asymptote at Po = 0.0005 (blue curves) (note: this reviewer also performed the analysis with Hill-transformed data, with exactly the same observations as with the simple Po graphs). The red curves correspond to the fit in the present version of the paper, and it is the “optimal” fit found by the program, probably because it minimizes residuals at 0.5 mM and 1 mM Ca^2+^. Yet, because the asymptotes are the important aspect of the graph, the Po values at the lowest Ca^2+^-concentrations are the relevant ones to consider. The blue fits are comparable visually and in terms of the residuals to the unconstrained fit, and they do better with the data points at both the lower and highest Ca^2+^-concentrations. The high quality of the blue fits clearly demonstrates that the data are also consistent with the *X_-_*and *X_+_* asymptotes having both the same temperature dependence. In fact, the value of *X_-_*is completely undeterminable by the data provided by the authors. In light of this analysis, the only conclusions that can be drawn are that the *X_+_* asymptote seems to have little temperature dependence. These limitations imposed by the data significantly limit the mechanistic insight that can be drawn, with the following outstanding conclusions in its present version: MthK channels exhibit a large step-like increase in Po between 32 and 37C. This increase is absent in a construct lacking the C-terminus of the channel, and the maximal open probability at saturating Ca^2+^ is only weakly temperature-dependent.

The authors may well be correct that temperature is affecting coupling between calcium binding and channel opening. However, that conclusion should be developed in a more nuanced fashion because the authors cannot measure the limiting Po at low calcium. Therefore, we suggest that the authors make an additional effort to soften the conclusions they develop in the section beginning "Coupling of RCK domains with the pore". They should specify that they cannot directly measure the limiting Po for either temperature, and they should describe how this is important. (Currently, it is not stated as a limitation but as an impossible goal). In the case of the Po-Ca relation at high temp, the slope is more shallow than at lower temperatures, making it likely that the Po at high temp and low Ca will plateau as their model predicts, but the plateau is not defined experimentally. They should also explicitly mention the implications about changes in slopes related to their model and conclusion.

---

## [Author Response]

Essential revisions:1) It appears that the bacterial growth-rescue experiments (Figure 1) were only carried out once. This raises concerns about their reproducibility. Information on experiment replication should be included. It appears that the authors simply performed densitometric measurements based on imaging of the cultures, which is not a robust means of quantification. Ideally, quantification should be performed by measuring OD600 of mini suspension cultures in triplicate for each condition. Also, it would be ideal to include a control with a MthK channel blocker, as it is possible that some of the growth deficiencies in truncated constructs result from temperature-dependent misfolding or other alternative mechanisms unrelated to potassium permeation through the channels.

We did not simply perform densitometric analysis but counted the number of colonies at different dilutions to obtain estimates of bacterial rescue. We have performed replicates but here to address any lingering concerns, we have added new experiments measuring the effect of temperature on bacterial growth rescue assay in suspension cultures. Growth was monitored by measuring OD600 values and these experiments were repeated at least three times for each condition (see Figure 1—figure supplement 1 legend).

Our data shows that MthK IR are more temperature sensitive compared with the MthK ΔC and empty vector (Figure 1C). Moreover, we also performed the experiments in the presence of blocker (5 mM barium) (Figure 1—figure supplement 1). In the blocker group, the growth difference among bacteria expressing various constructs is diminished between 37 °C and 24°C. These results suggest that higher MthK activity at 37 °C compared to 24 °C is responsible for better complementation in bacterial growth.

We should note that while the results of the new experiments agree with our original plate assays, the differences between the two conditions (24 and 37 °C) is not as dramatic in the suspension culture compared to the plate assay. We do not fully understand why that is the case and may simply reflect the differences in metabolic rate or non-linearity of optical density measurements at high values. Nonetheless, our new assays also show that the MthK IR has a higher growth rate at 37 °C compared to 24 °C.

Details are also included in the main text subsection “MthK expressed in *E. coli* is temperature-sensitive”.

2) The authors should provide data showing that the steep temperature-dependent increase in Po that occurs between 32 and 37C is reversible, showing that once channels have reached maximal activation, Po returns back to the same level it had at the beginning of the experiment at room temperature. All analysis and mechanistic interpretations assume that everything happens at equilibrium, but no evidence is provided to support this key assumption. Given that temperature-activated TRP channels exhibit prominent hysteresis when activated by heating, it is also important to show that the same Po at 37C degrees can be consistently achieved in repeated stimulations in the same patch. If this were not the case, then the thermodynamic analysis performed on the data would not be valid. Notably, MthK was chosen as a model system because eukaryotic temperature-activated channels are polymodal and their responses to temperature often involve out-of-equilibrium processes. Yet, no data is provided showing that MthK – which is also a polymodal receptor – is exempt from these same issues.

These are valid concerns. We should note that repeated stimulations back forth between two temperatures in same patch is just not possible. The TRPV1 experiments were macroscopic measurements whereas with single channel measurement, one needs to collect data for sufficient length of time. This problem especially acute when we have low Po values as in low temperatures. Nonetheless, we have carried out one cycle of low-high-low temperature experiment on multiple patches. Our results show that after maximal activation of MthK by high temperature, upon returning back to the room temperature the Po returns to the same level it had at room temperature (n = 6 independent recordings) (Figure 2—figure supplement 3).

Also, our minutes long recordings at high-temperature show no evidence of loss of activity (Figure 2—figure supplement 1).

This discussion is also included in our main text subsection “Coupling of RCK domains with the pore”.

3) In Figure 2 we only see the Po increases at 1 temperature value, 39 degrees, while for all other values tested, there is no effect, which is somewhat odd, and also at odds with the more monotonic increase in Po with temperature in Figure 1—figure supplement 1E. Is this real and is it happening in all experiments? The authors should discuss this.

As suggested by the reviewers, we have repeated the temperature step experiments at 0.1 mM Ca^2+^ with single channel recordings with independent patches from n = 5 (21 °C), 4 (26 °C), 4 (32 °C), 5 (37 °C). As shown in the Figure 2D, the dramatic increase in Po only happens between 32 °C and 37 °C in single channel recordings.

The differences between macroscopic measurements vs single channel measurements are discussed next.

4) The temperature-dependence of the macroscopic currents in Figure 1—figure supplement 1 appears negligible. Given the enormous fold-differences in rescue between room temperature and 36C in Figure 1, one would expect to see similar steeply temperature-dependent changes in the macroscopic currents if the rescue is indeed reflecting changes in potassium conductance, that should be evident even if incomplete Ba^2+^-block results in over-subtraction of currents at higher temperatures. Why were experiments not carried out at lower Ca^2+^-concentrations (or in its absence), where temperature-dependence might be more pronounced? The authors should also show recordings and corresponding group data for spheroplasts that are not expressing MthK channels at different temperatures and Ca^2+^-concentrations. The authors should discuss the possibility that temperature-dependence of MthK channels is dependent on the expression system and possibly the lipid environment. Is it possible that the difference in bacterial strains used in the rescue experiments vs the macroscopic current recordings could be responsible for the apparent lack of temperature sensitivity in the latter?

We respectfully disagree with the comment that the temperature-dependence of macroscopic currents is negligible. If you compare the currents with and without barium block at low temperature, it is evident that all the observed macroscopic current is leak at that point. At higher temperatures, MthK currents above the leak currents become obvious. We cannot use the low temperature value (which is essentially zero) for Q_10_ calculation, therefore in the original version we calculated Q_10_ by using the slope. However, the estimate of Q_10_ using slope is meaningful only when we are able to measure temperature dependent increase in currents over the whole range and it is evident that in spheroplasts currents do not saturate at 41 °C. We apologize for that oversight and in the revised version, we have modified the text. See subsection “MthK expressed in *E. coli* is temperature-sensitive”.

We agree with the reviewers that there are some differences between spheroplasts recordings and reconstituted systems especially with regards to their calcium dependence etc. But trying to resolve the underlying reasons for these differences is outside the scope of the current work.

5) With the single-channel conductance in Figure 2, the authors show a graph (C) where conductance goes up with temperature, with a low value at 21 degrees, but this is not visible from traces in A, or from those in Figure 2—figure supplement 1. But these all look identical. This issue should be revisited.

The reviewer is correct. After we added more datasets and plotted the conductance values it is clear that the change is not statistically significant (Figure 2C).

6) A serious concern is the apparent lack of independent replicates in the single-channel data. For example, it is clearly stated that all traces in Figure 2A are from one patch but it is not stated how many patches were recorded, what is the average behavior of these individual recordings. There is no plot where a mean value with errors from recordings in different patches is shown. Has this been done for more than one patch? Graphs in C and D should show averages over multiple patches, but they do not. The open time plot for example is actually a mean open time from just one experiment with many events. For the bar graph in D, the authors say: "error bars are SEM calculated by bootstrapping from 10 sweeps of 30 s length". This is just an n of 1 and needs to be repeated.We have the same concern for the data shown in Figures 3 and 4. In Figure 3, only one experiment is analyzed (performed?) for the pore-only MthK channel. In Figure 4, it appears that the two Po vs Ca^2+^ plots in C are derived from just one experiment at each Ca^2+^ concentration, where temperature was changed from 21 to 37C. Please clarify if this is an n of 1 for each calcium concentration, in which case it needs to be repeated. If not, the number of separate patches should be clearly stated in the figure legend.For all of these experiments, the authors must clearly indicate the number of patches from which data was collected for each condition and channel type, clearly denoting which represent independent replicates, and if they are biological or technical replicates. It is essential that the main findings in Figure 3 and Figure 4C are supported by at least three independent replicates, ideally more.

This is major concern and we absolutely agree with the reviewers that we should add more replicates to make the data statistically robust. As described below, we have added independent replicates for each condition.

Error bars for Figure 2C and 2D are SEM calculated from n = 5 (21 °C), 4 (26 °C), 4 (32 °C), 5 (37 °C) independent measurements.

Error bars in Figure 3C represents SEM calculated from n = 5 independent patches.

Error bars in Figure 4C and Figure 4—figure supplement 1 represent SEM calculated from n = 5 (0.1 mM Ca^2+^, 21 °C), 3 (0.2 mM Ca^2+^, 21 °C), 5 (0.5 mM Ca^2+^, 21 °C), 6 (1 mM Ca^2+^, 21 °C), 4 (2 mM Ca^2+^, 21 °C), 4 (5 mM Ca^2+^, 21 °C), 5 (10 mM Ca^2+^, 21 °C), 4 (0.1 mM Ca^2+^, 37 °C), 3 (0.2 mM Ca^2+^, 37 °C), 4 (0.5 mM Ca^2+^, 37 °C), 3 (1 mM Ca^2+^, 37 °C), 4 (2 mM Ca^2+^, 37 °C), 3 (5 mM Ca^2+^, 37 °C), and 3 (10 mM Ca^2+^, 37 °C) independent recordings.

All these information is also clearly indicated in the corresponding figure legends.

7) For the NPo measurements in Figure 3A; based on the recordings shown, it is difficult to see how the data at 36C (red trace) have a 4-fold lower NPo than the data at 20C (blue trace), unless the red trace includes some very long inactive periods. The authors should explain this further.

NPo is not just a function of number of openings but also the length of the open dwell times. Although the number of openings at 20 °C and 36 °C are not too different, the dwell times at low temperature is four-fold longer than at 36 C (see insets in Figure 3A). Therefore, this difference in dwell time accounts for all the differences between NPo at two temperatures.

8) It would be useful for the readers that the authors discuss their proposed mechanism of how temperature increases Po from the perspective of the existing structures of MthK, rather than only abstractly. Especially since the mechanism invoked by the authors involves linkage between the ligand-binding domain and the pore domain and these linkers are actually resolved in the closed channel structure. The authors should not use "helices" to describe the linker, since it is only a short loop that connects the end of the TM2 helix with the RCK domain, at least in the closed state structure, where it is visible.

As suggested by the reviewers, we have further illustrated our proposed mechanism from the new closed structure of MthK in the Discussion. The structure of closed MthK channel shows that the linker is more stable without Ca^2+^ versus Ca^2+^ activation state of RCK. The stability is a result of hydrophobic interactions as well as salt bridges between linker and RCK. Similarly, we think temperature may affect these interactions to modify the coupling between RCK domain and the pore (Fan et al. (2020) *Nature* 580(7802), 288-293). This is also included in Discussion.

We have now removed the word “helices” when describing the linker.

[Editors' note: further revisions were suggested prior to acceptance, as described below.]

Revisions for this paper:1) The authors do not provide enough evidence to strongly conclude that the pore domain has no contribution to temperature sensing. First, in the complementation assays, there is no evidence that δ-C forms viable channels. These data should be removed. Second, there are dramatic temperature-dependent changes in the gating of single channels lacking the C-terminus – the mean closed dwell times are noticeably shorter (see Figure 3A and Figure 3—figure supplement 1C), which is what largely contributes to a temperature-dependent change in Po for the constructs with an intact C-terminus. It is thus undeniable that the construct without a C-terminus undergoes temperature-dependent conformational transitions – it might just be that these don't lead to an increase in Po due to the large disruption in the overall protein energy landscape caused by the deletion of a functionally important part of the channel. Finally, the range of high temperatures explored by the authors is narrow, particularly compared to the ranges over which heat-sensitive TRP channels are activated, which is above the 36 C (or 39 C, although these data were not included in the manuscript). Whereas it is true that this limitation applies to all other constructs, it is also true that all other constructs did show temperature-dependent changes in Po below 39 C. This point is central to the conclusions in the manuscript and must therefore be discussed more factually. The conclusions regarding the MthK-PO construct should be toned down throughout the manuscript.

First, let us consider the predictions of linkage analysis. If the pore gating is intrinsically sensitive to temperature, then the *ln(Po/Pc)* vs. calcium curves will be displaced equally at both high and low calcium concentrations. This is not a parameter dependent conclusion but comes directly from the linkage equations (See Equation 3A and 3B). We are not aware of any scenario where this would not be true (see also the derivation of general model with N binding sites in Materials and methods). Our data clearly shows that at saturating calcium concentrations, the *ln(Po/Pc)* values are hardly affected by temperature in contrast to those at low calcium concentrations. Second, our single channel data of the MthK PO construct shows that increasing the temperature actually decreases the Po of the channel at high temperature by four-fold. The dependence is opposite to what one would expect if the temperature-dependence of the pore domain had major contribution to thermosensitivity of MthK. Third, with regards to the temperature-range, we are not aware of a single study where TRPV1 (or any thermosensitive channel) temperature-sensitivity was characterized using single channel recordings at significantly higher temperature (>42 °C). All the high-temperature recordings for TRPV1 and other channels are macroscopic recordings (typically temperature ramps) which do not require prolonged exposure of the patch to high temperatures. Unfortunately, for meaningful single channel recordings we need to have stable patch for at least 30-40 minutes or more. We find that the patch stability decreases dramatically as one goes higher than 40 °C and see no way to address this question using current methods. Moreover, this question can never be answered because there is always a possibility that MthK PO construct is thermosensitive beyond experimental range. Finally, with regards to the complementation assay, the plate assays at 28 and 24 °C clearly shows that the IPTG induction of the MthK DC construct is 100 fold more effective at growth rescue in low potassium (Figure 1D) and therefore must form viable channels.

We have now discussed these points in a separate paragraph in the Discussion. In addition, we have also clarified the implications of the linkage equations to highlight this point.

2) The Hill plots in Figure 4C do not allow the authors to accurately determine the value of the asymptote at the lower Ca^2+^ concentrations (X-) and thus the value of deltaX is also equally undetermined; there is over an order of magnitude change in the value of the Hill plot from the lower Ca^2+^ concentration to the second lowest at both temperatures, arguing against a plateauing over the examined range of concentrations. Without this, no quantitative analysis of the data can be obtained. Yet, the data do show that qualitatively the lower asymptote, wherever it might fall, is more temperature-dependent than the upper asymptote, which still supports the authors conclusions. Therefore, the argument must be framed in a qualitative manner. (We note that you provide estimates of coupling energies in the context of "classical allosteric analysis", citing Horrigan and Aldrich, but we didn't see that type of analysis presented in this manuscript. We also note that your results are quantitatively different from those reported by Thomson and Rothberg, 2010. Framing your results in a qualitative rather than quantitative way will resolve these issues.

We agree with the reviewers but want to add for the record that we have more confidence in the value at 0.1 mM calcium asymptote (X-) because these data were obtained from 10 independent patch recordings. The value at 0.2 mM was obtained from 3 independent recordings. Therefore, using the value at 0.1 mM calcium, we are able to place limits on coupling energy at 21 ºC. In the worst-case scenario, the coupling energy between the calcium sensor and pore is going to be higher if the curve saturates at even lower values. This is completely consistent with our message that the coupling energy changes with temperature as the reviewers have pointed out. In accordance to their wishes, we have eliminated the paragraph comparing our energy values with the Horrigan and Aldrich studies.

With regards to Thomson and Rothberg, 2010 paper, we would just like to add that our measurements were all obtained using liposome patch and those measurements were obtained in BLMs. The functional behavior of the MthK channels appears to be somewhat different between the bilayers and patch recordings. For instance, MthK channels do not seem to inactivate in bilayer recordings (personal communications Rothberg and Nimigean Lab).

We suggest you could include one paragraph where you examine how the theoretical framework (please elaborate) applies specifically to your experimental data. You could test whether an X_-_ asymptote can truly be inferred from the data (perhaps comparing between fits with different limiting values). We would like you to provide a compelling argument that no other model (for example, one with a temperature-dependent Ca^2+^-association constant – see Figure 5—figure supplement 1B) is consistent with your data. It would strengthen the paper if you can show that your Hill plot data at the two temperatures cannot be described by two curves that have the same X_-_ asymptote at a much lower ln(Po/(1-Po)) value than those observed in the data.

Done. We discuss the simple situation where the single binding site is temperature dependent or a more complicated scenario where the MthK is regulated by two binding site- one is temperature-dependent whereas other is not. We have added a new figure to make this point (Figure 5—figure supplement 2).

Revisions expected in follow-up work:Experiments essential to support the conclusion that the pore domain has no contribution to temperature sensing are as follows. First, the complementation assays should include western blots showing that the expression of the construct lacking the C-terminus is not heavily compromised in the context of this assay. Without these data, it is not possible to conclude that the lack of rescue by this construct is due to an absence of temperature-dependent activation, particularly because that construct also fails to rescue at room temperature.

See our response to point #1. Western blots will not rule out the possibility that the MthK ΔC construct is non-functional. The ultimate test is to show that the isolated purified channel without the RCK domain is not sensitive to temperature which we have done.

Second, a higher range of temperatures should be explored. In the TRP channel literature, it is standard to go at least over 40 C, because below those temperatures no steeply temperature dependent responses are usually observed; MthK comes from a thermophilic organism that grows optimally between 55-60C, so a lack of response at 37C is not sufficient to conclude there is no temperature response.

Thank you for the suggestion. There are two things consider here. First, as far as we are aware, there is no study where single channel recordings were obtained for extended periods at temperatures above 40 ºC. For characterization of low open probabilities and proper estimate of number of channels in a patch, we need patches to hold for at least 30-40 minutes. These measurements are extremely challenging and the fact that there is no other comparable study in the literature speaks for itself. Recordings at high temperatures are made with macroscopic currents which are easier to obtain but have other issues. For instance, one has to properly subtract the leak currents since leak is temperature-dependent. Most studies on TRPV1 and other heat-sensing channel using temperature ramps are unfortunately non-equilibrium measurements. Second, we have to consider how much more change can we expect. The Po values at 21 ºC in low calcium (where the temperature-sensitivity is maximum) is already at 0.2 at 37 ºC. The maximum it can reach is 1 which is only 5 fold change. We would of course want to obtain these measurements at slightly higher temperatures than described here but it does not seem we have to go much further to saturate the temperature-response despite the fact that archaebacteria grows in 55 -60 ºC.

[Editors' note: further revisions were suggested prior to acceptance, as described below.]

Revisions:The model simulations in Figure 5 and Figure 5—figure supplements 1-3 are adequate for a review article on linkage analysis, but are not useful for analyzing the experimental data that the paper is about; they are all graphed at distinct scales than the data in Figure 4C, and it is impossible to determine which of those model predictions actually describes the data accurately and which ones don't.

The simulations shown here are meant for illustration only and it is not our intention to carry out an exercise of model fitting. We want the readers to focus on the fact that these are model-agnostic interpretations. If any of the model parameters are temperature-sensitive, one can predict the behavior of the asymptotes by taking derivative of *X_-_*(Equation 3A and 21) and *X_+_* (Equation 3B and 22) with respect to temperature. Furthermore, in subsection “Coupling of RCK domains with the pore”, we have also qualitatively explained how to interpret the temperature dependence of these asymptotes.

Using the data provided by the authors (Figure 4—figure supplement 1), one of the reviewers fit the Po vs [Ca^2+^] relations with the Hill equation used by the authors without applying any constraints on the fitting (red curves in the graph) and by constraining fits at both 21 and 37C to have the same X_-_ asymptote at Po = 0.0005 (blue curves) (note: this reviewer also performed the analysis with Hill-transformed data, with exactly the same observations as with the simple Po graphs). The red curves correspond to the fit in the present version of the paper, and it is the “optimal” fit found by the program, probably because it minimizes residuals at 0.5 mM and 1 mM Ca^2+^. Yet, because the asymptotes are the important aspect of the graph, the Po values at the lowest Ca^2+^-concentrations are the relevant ones to consider. The blue fits are comparable visually and in terms of the residuals to the unconstrained fit, and they do better with the data points at both the lower and highest Ca^2+^-concentrations. The high quality of the blue fits clearly demonstrates that the data are also consistent with the X_-_ and X+ asymptotes having both the same temperature dependence. In fact, the value of X_-_ is completely undeterminable by the data provided by the authors. In light of this analysis, the only conclusions that can be drawn are that the X+ asymptote seems to have little temperature dependence. These limitations imposed by the data significantly limit the mechanistic insight that can be drawn, with the following outstanding conclusions in its present version: MthK channels exhibit a large step-like increase in Po between 32 and 37C. This increase is absent in a construct lacking the C-terminus of the channel, and the maximal open probability at saturating Ca^2+^ is only weakly temperature-dependent.

Before we get into rebutting this point specifically, we would like to point out that in the previous round, the reviewers stated that “the data do show that qualitatively the lower asymptote (is) more temperature-dependent than the upper asymptote, which still supports the authors conclusions.” These statements are meaningless if we keep revisiting the same topic over and over again.

We have a lot of issues with post-hoc analysis of the Po vs. calcium data by the reviewer.

First, fitting the Hill equation to semi log transformation is problematic because of weighting of residuals. In a log transform, data points which are less reliable are given more weight than in linear fitting which leads to increased errors in parameter estimation (see Dowd and Riggs (1965) JBC 240, 863-9 for extensive discussion about this in enzyme kinetics parameter estimation).

Second, the logic that the residuals should be ignored and the data should be constrained to Po values determined at 0.1 mM escapes us. By extending your logic, we can also fit the same data to a straight line, which implies non-specific binding and predicts that the Po values will be greater than 1 at higher calcium concentration!!

Third, let us consider the scenario that there is a higher affinity calcium binding site whose affinity is much greater than 0.1 mM and therefore the observed effects at low calcium (0.1 mM) is dominated by the temperature-dependence of that site. This is a pertinent concern even if we had multiple datapoints exhibiting apparent saturation of Po values. In particular, this is also an issue for stimuli such as pH, voltage and temperature where “zero stimulus” condition is not experimentally accessible.

To simplify the analysis, let us ignore the low affinity calcium binding site since it is not temperature-dependent in this model and therefore will be identical at both temperatures. If we now focus on the high-affinity site, the lower asymptotes are unchanged (by definition) but one must account for the differences in Po for the two curves at 0.1 mM. The only possibility is that the upper asymptotes for the high-affinity calcium site is different and is the primary contributor to temperature-dependence observed at 0.1 mM. Based on the linkage equations, asymmetric effects on the asymptotes are observed only and only when the coupling between the calcium sensor and the pore gating is temperature-dependent. The difference is that now the coupling between the bound sensor and pore is temperature-sensitive as opposed to our findings on the low affinity site where the coupling between the apo sensor and pore is temperature-dependent. In the end, however, both these models have to invoke a change in coupling between calcium sensor and pore gates to account for the observed temperature dependence.

The authors may well be correct that temperature is affecting coupling between calcium binding and channel opening. However, that conclusion should be developed in a more nuanced fashion because the authors cannot measure the limiting Po at low calcium. Therefore, we suggest that the authors make an additional effort to soften the conclusions they develop in the section beginning "Coupling of RCK domains with the pore". They should specify that they cannot directly measure the limiting Po for either temperature, and they should describe how this is important. (Currently, it is not stated as a limitation but as an impossible goal). In the case of the Po-Ca relation at high temp, the slope is more shallow than at lower temperatures, making it likely that the Po at high temp and low Ca will plateau as their model predicts, but the plateau is not defined experimentally. They should also explicitly mention the implications about changes in slopes related to their model and conclusion.

Please see our response to point #2 where we consider the possibility that the limiting Po is not experimentally determined.

We have added the following sentences at the end of the Results section.

“We can draw a similar conclusion by comparing the Hill coefficient of the dose response curves at two temperatures. Under certain conditions, the Hill slope of a response curve is a measure of cooperativity (60) and the fact that the calcium dose-response curve at 37 °C has a lower Hill coefficient than those obtained at 21 °C further supports the notion that coupling interaction between various elements is reduced at higher temperatures in MthK.”